# Phytoalexin sakuranetin attenuates endocytosis and enhances resistance to rice blast

Lihui Jiang[1,8], Xiaoyan Zhang[1,8], Yiting Zhao[1,2,8], Haiyan Zhu[1], Qijing Fu[1], Xinqi Lu[1], Wuying Huang[1], Xinyue Yang[1], Xuan Zhou[1], Lixia Wu[1], Ao Yang[1], Xie He[1], Man Dong[1], Ziai Peng[1], Jing Yang[1,3,4], Liwei Guo[1,3,4], Jiancheng Wen[5], Huichuan Huang [1,3,4], Yong Xie[1,3,4], Shusheng Zhu[1,3,4], Chengyun Li[1,3,4], Xiahong He[6], Youyong Zhu[1,3,4], Jiří Friml[7] & Yunlong Du[1,3,4] ✉

Phytoalexin sakuranetin functions in resistance against rice blast. However, the mechanisms underlying the effects of sakuranetin remains elusive. Here, we report that rice lines expressing resistance (R) genes were found to contain high levels of sakuranetin, which correlates with attenuated endocytic trafficking of plasma membrane (PM) proteins. Exogenous and endogenous sakuranetin attenuates the endocytosis of various PM proteins and the fungal effector PWL2. Moreover, accumulation of the avirulence protein AvrCO39, resulting from uptake into rice cells by *Magnaporthe oryzae*, was reduced following treatment with sakuranetin. Pharmacological manipulation of clathrin-mediated endocytic (CME) suggests that this pathway is targeted by sakuranetin. Indeed, attenuation of CME by sakuranetin is sufficient to convey resistance against rice blast. Our data reveals a mechanism of rice against *M. oryzae* by increasing sakuranetin levels and repressing the CME of pathogen effectors, which is distinct from the action of many R genes that mainly function by modulating transcription.

The plant immune system is essential to plant survival and adaptation to the environment. In rice (*Oryza sativa*), rice blast is one of the major diseases and is caused by the fungus *Magnaporthe oryzae*. Many R genes have been discovered to confer resistance to rice blast[1–11]. Of these resistance genes, *Pi9* confers broad-spectrum resistance against *M. oryzae*[8]. *Pi9* expression in rice near-isogenic lines (NILs) resistant to *M. oryzae* modulates genome-wide transcription, enriching gene products involved in both primary and secondary metabolism[12]. Some effectors, such as PWL2, are translocated into the rice cell cytoplasm

via CME[13,14], where certain R genes can recognize effectors of *M. oryzae* and activate rice resistance pathways[15]. However, little is known of the mechanisms by which essential R genes signaling pathways mediate rice resistance to *M. oryzae*.

Endocytosis is an important cellular event in plant defenses against disease[16,17], and inhibition of endocytosis can attenuate the induction of plant defense responses[18,19]. CME is involved in plant innate immunity[20], and effectors from fungi[21], insects[22] and oomycetes[23] can enter the plant cytoplasm through endocytosis.

[1]College of Plant Protection, Yunnan Agricultural University, Kunming 650201, China. [2]Shanxi Agricultural University/Shanxi Academy of Agricultural Sciences. The Industrial Crop Institute, Fenyang 032200, China. [3]State Key Laboratory for Conservation and Utilization of Bio-Resources in Yunnan, Yunnan Agricultural University, Kunming 650201, China. [4]Key Laboratory of Agro-Biodiversity and Pest Management of Education Ministry of China, Yunnan Agricultural University, Kunming 650201, China. [5]Rice Research Institute, Yunnan Agricultural University, Kunming 650201, China. [6]Key Laboratory for Forest Resources Conservation and Utilization in the Southwest Mountains of China, Ministry of Education, Southwest Forestry University, Kunming 650224, China. [7]Institute of Science and Technology Austria (IST Austria), Klosterneuburg, Austria. [8]These authors contributed equally: Lihui Jiang, Xiaoyan Zhang, Yiting Zhao. ✉e-mail: yunlongdu@aliyun.com

Membrane-localized resistance proteins are known to mediate effector-triggered immunity (ETI), possibly by regulating endocytosis of the immune receptor complex[24], however, whether R genes are able to regulate the endocytosis of effectors to induce plant defense responses remains unclear.

Phytoalexins are secondary metabolites produced in plants in response to pathogen attack and environmental stresses. Sakuranetin is a major phenolic phytoalexin, and shows strong antimicrobial activity against fungal pathogens, including *M. oryzae*[25]. Sakuranetin biosynthesis requires NOMT[26,27] as well as the jasmonic acid (JA)-mediated signaling pathway[28–30]. However, the mechanism by which sakuranetin confers rice resistance to *M. oryzae* at the cellular level remains unclear.

In this study, we found that rice lines containing R genes showed higher induced levels of sakuranetin and attenuated cellular endocytosis following infection with the fungus *M. oryzae*. Furthermore, sakuranetin decreased endocytosis of fungal effectors in rice root cells via a clathrin-mediated pathway. These results revealed an unsuspected mechanism by which the phytoalexin sakuranetin may enhance plant resistance by down-regulating cellular endocytosis.

## Results

### Rice lines resistant to rice blast show attenuated endocytosis

Given that endocytosis is prominently involved in pathogen defense, we investigated its potential link to rice blast resistance by comparing resistant and non-resistant rice lines. We obtained several rice NILs, including IRBLa-A, IRBL19-A, IRBLta-CP1, IRBL20-IR24, IRBLi-F5, IRBL3-CP4, IRBLk-K, IRBLsh-S, IRBLKs-S and IRBL9-W containing the R genes *Pia*, *Pi19*, *Pita*, *Pi-2O*, *Pii*, *Pi3*, *Pi-k*, *Pish*, *PiK-s* and *Pi9*, respectively. We then confirmed the resistance of these lines to rice blast by inoculating them with the Guy11 strain of *M. oryzae* (Fig. 1a–k). Compared with the control rice line Lijiangxintuanheigu (LTH), which is susceptible to *M. oryzae*[31], these rice NILs all showed significant resistance to rice blast (Fig. 1l), and all showed higher levels of sakuranetin in their leaves (Supplementary Fig. 1).

Next, we investigated endocytosis in these rice NILs. Cells were stained with FM4-64, the established endocytic fluorescent tracer dye, which is incorporated to the PM and can be internalized only by endocytosis[32]. Compared with the LTH control (Fig. 1m), the cytoplasm fluorescence was significantly decreased in the root epidermal cells of all the tested rice NILs (Fig. 1n–x).

We then further analyzed these endocytosis in NILs following treatment of the roots with the protein trafficking inhibitor brefeldin A (BFA). We found that the FM4-64-labeled BFA bodies (the BFA-induced aggregations of endosomes) were both smaller and fewer in number in the roots of all the tested NILs compared to the control LTH roots, and that this reduction was correlated with higher levels of sakuranetin (Supplementary Fig. 2a–n). This demonstrates that the NILs show a lower rate of plasma membrane internalization, and thus confirms the attenuated endocytosis in these NILs.

*Pigm* is an R gene conferring resistance to rice blast[11]. Twenty minutes following FM4-64 labeling, the PM fluorescence intensity of the rice line Kongyu131 expressing *Pigm* (KY131-*Pigm*) (Supplementary Fig. 3b, c) was enhanced compared to the wild-type KY131 (Supplementary Fig. 3a). Moreover, roots of the rice line KY131-*Pigm* (Supplementary Fig. 3e–g) treated with BFA and labeled with FM4-64 also showed smaller and fewer BFA bodies compared to the wild-type KY131 (Supplementary Fig. 3d, f, g). Furthermore, the roots of rice line KY131-*Pigm* had higher sakuranetin levels than did those of the wild-type KY131 (Supplementary Fig. 3h).

We then checked the PM fluorescence intensity of several other rice lines that are susceptible to rice blast, including *ipa1*[33] (Supplementary Fig. 4a, b), *NahG*[34], *OsNPR1*-RNAi[35], *arf12*[36], *OsROD1*–over-expression line[37] and *35S::miR393a* line[38]. Compared to the wild-type, the PM fluorescence intensity of the root epidermal cells in all of these

rice seedlings was decreased following FM4-64 labeling (Supplementary Fig. 5a–k), and this decrease was shown in all cases by lower sakuranetin levels than the control (Supplementary Fig. 5l). Thus, our data from many rice lines either resistant or susceptible to rice blast demonstrate that resistance to rice blast, rate of endocytosis and sakuranetin levels are strongly correlated.

### Attenuation of endocytosis by sakuranetin did not interfere with endosomal dynamics and PM fluidity

We further analyzed the effect of sakuranetin on endocytosis by treating the LTH and Nipponbare (NPB) wild-type lines with sakuranetin. As compared to the mock-treated control (Fig. 2a, c), the fluorescence intensity of the plasma membrane FM4-64 staining was significantly increased following sakuranetin treatment (Fig. 2b, d, e). When the rice wild-type LTH and NPB lines were treated with BFA together with FM4-64 labeling, BFA bodies in the root epidermal cells were both smaller and reduced in number in the roots treated with sakuranetin (Fig. 2g, i–k) than in the mock-treated controls (Fig. 2f, h, j, k).

To test whether the attenuation of endocytosis in rice root cells is a dose-response of sakuranetin treatment, rice roots were co-treated with 10, 25 and 50 μM sakuranetin and 25 μM BFA (Supplementary Fig. 6b–d). Compared with the control (Supplementary Fig. 6a), the observed decreases in size and number of BFA bodies correlated with the increases in sakuranetin concentration (Supplementary Fig. 6e, f). This demonstrates that sakuranetin is able to decrease endocytosis in rice root cells in a dose-dependent manner.

To provide an alternative to FM4-64 staining, we looked at the BFA-sensitive trafficking of PIN auxin transporters[39] in the rice NPB lines expressing *ProOsPIN1b::OsPIN1b-GFP*, *ProOsPIN2::OsPIN2-GFP* and *35S::OsPIN3t-GFP*. As before, when treated with sakuranetin together with BFA (Supplementary Fig. 7b, d, f, g), smaller GFP fluorescence-visualized BFA bodies were observed compared to the controls treated with BFA alone (Supplementary Fig. 7a, c, e, g). Next, we examined the effect of sakuranetin treatment on rice protoplast cells expressing the rice syntaxin of plants 121 (*OsSYP121*), which localizes in the PM and is involved in resistance to rice blast[40]. We found that sakuranetin treatment enhanced the localization of OsSYP121-GFP to the plasma membrane compared to the mock-treated controls (Supplementary Fig. 8). These data show that the phytoalexin sakuranetin attenuates endocytosis and BFA-sensitive endocytic trafficking in rice.

We next examined the ultrastructure of organelle and vesicle compartments in the roots of rice wild-type NPB treated with sakuranetin, and found that there were no obvious changes (Supplementary Fig. 9b) compared with the control treated with DMSO (Supplementary Fig. 9a). Moreover, there were no differences in the organelle and vesicle compartments between the rice subjected to the BFA and sakuranetin/BFA treatments, and the BFA bodies could still be observed using electron microscopy in the root epidermal cells treated with sakuranetin (Supplementary Fig. 9c, d). To test whether sakuranetin interferes with the role of BFA in intracellular dynamics, we examined the effect of sakuranetin on BFA-induced aggregations of *trans*-Golgi network/early endosome (TGN/EE) markers. We used the OsSYP121-GFP to label TGN/EE[41]. In rice protoplast cells expressing *OsSYP121-GFP*, the BFA alone or BFA together with sakuranetin treatment caused aggregations of the marker (Supplementary Fig. 9e, f). However, sakuranetin did not affect the mobility of the multivesicular body/prevacuolar compartment (MVB/PVC) labeled with ras-related protein RABF2a (ARA7)[42,43] fused to GFP (Supplementary Fig. 10a–c), the endoplasmic reticulum (ER) labeled with the ER retention signal protein (HDEL)[44] fused to GFP (Supplementary Fig. 10d–f) or the TGN/EE labeled by OsSYP121-GFP (Supplementary Fig. 10g–i). This reveals that sakuranetin did not generally interfere with endosomal dynamics.

We then constructed rice lines overexpressing *OsNOMT* (Supplementary Fig. 11a, b) to increase sakuranetin production

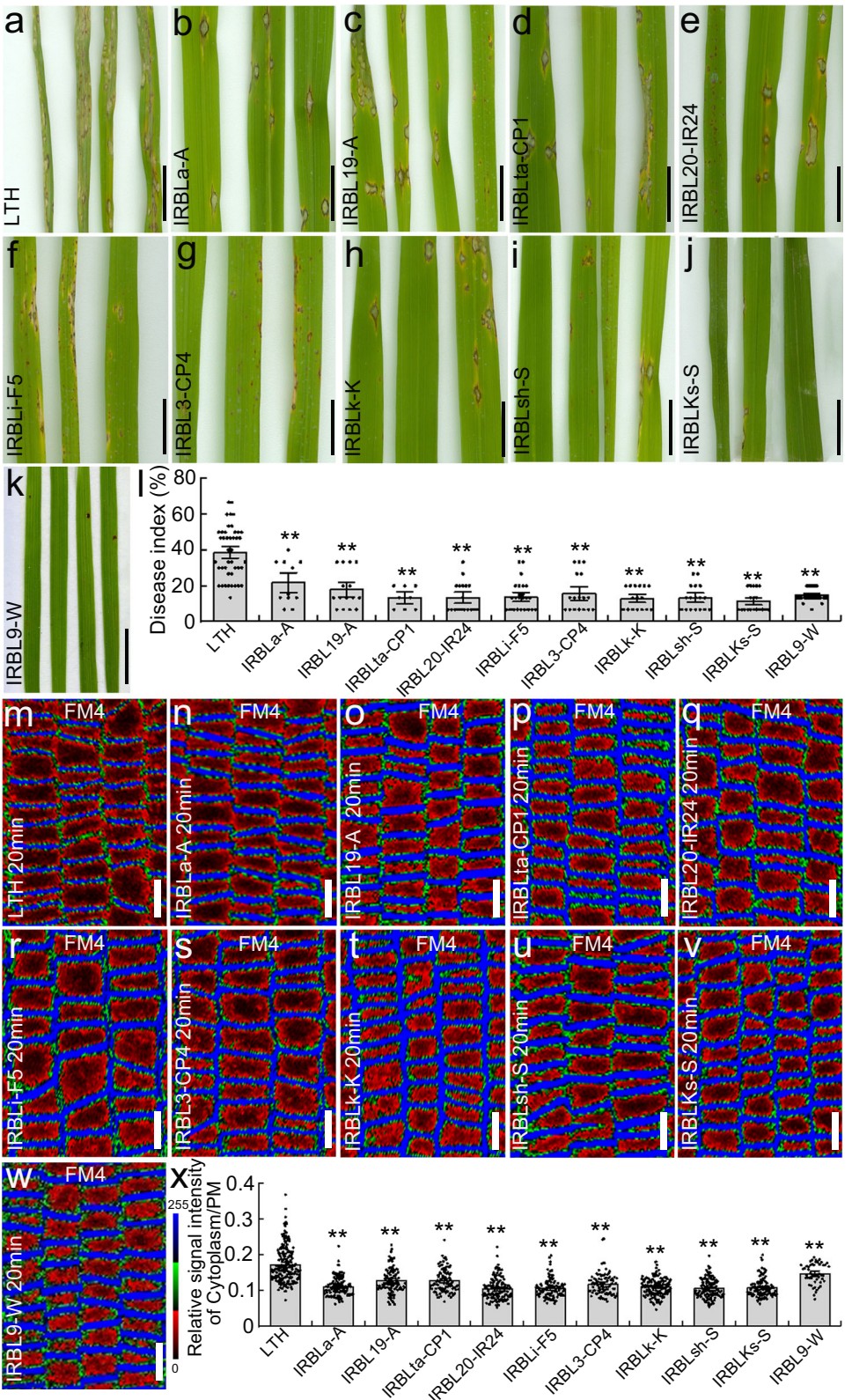

endogenously (Supplementary Fig. 11c, d). The size and number of FM4-64-stained BFA bodies in rice root cells were both obviously reduced in the lines overexpressing *OsNOMT* (#5, #7, #8, #12, #14, #24) (Fig. 2m–t) compared to the control wild-type rice NPB (Fig. 2l). In contrast, when we investigated endocytosis in rice *nomt*-cc mutants (in an IRBL9-W background) (Supplementary Fig. 12a, b), which showed reduced sakuranetin levels (Supplementary

Fig. 12c, d), the size and number of BFA bodies significantly increased in the *nomt*-cc mutant lines #1, #6 and #10 (Fig. 2v–z) compared with the IRBL9-W control (Fig. 2u). This shows that endogenous manipulation of sakuranetin biosynthesis also has an impact on endocytosis.

Sakuranetin is a lipophilic compound that has been shown to be incorporated in cell membranes[45]. To test whether sakuranetin

**Fig. 1 | The rice near-isogenic lines containing R genes show attenuated endocytosis and increased resistance to rice blast.** Disease lesion phenotype of *M. oryzae* on the rice line LTH (**a**), and the near-isogenic lines IRBLa-A (**b**), IRBL19-A (**c**), IRBLta-CP1 (**d**), IRBL20-IR24 (**e**), IRBLi-F5 (**f**), IRBL3-CP4 (**g**), IRBLk-K (**h**), IRBLsh-S (**i**), IRBLKs-S (**j**) and IRBL9-W (**k**). All rice lines were inoculated with the *M. oryzae* strain Guy11. **l** Quantification of the disease index shown in the images **a–k** ($n_{LTH}$ = 43, $n_{IRBLa-A}$ = 11, $n_{IRBL19-A}$ = 15, $n_{IRBLta-CP1}$ = 8, $n_{IRBL20-IR24}$ = 18, $n_{IRBLi-F5}$ = 24, $n_{IRBL3-CP4}$ = 17, $n_{IRBLk-K}$ = 16, $n_{IRBLsh-S}$ = 17, $n_{IRBLKs-S}$ = 19, $n_{IRBL9-W}$ = 27). Root epidermal cells of 6-day old rice LTH (**m**), and near-isogenic lines IRBLa-A (**n**), IRBL19-A (**o**), IRBLta-CP1 (**p**), IRBL20-IR24 (**q**), IRBLi-F5 (**r**), IRBL3-CP4 (**s**),

IRBLk-K (**t**), IRBLsh-S (**u**), IRBLKs-S (**v**) and IRBL9-W(**w**) were labeled with 4 μM FM4-64 for 90 min. **x** Quantification of relative fluorescence intensities of plasma membrane versus cytoplasm in rice root epidermal cells shown in the images **m–w** ($n_{LTH}$ = 194, $n_{IRBLa-A}$ = 127, $n_{IRBL19-A}$ = 121, $n_{IRBLta-CP1}$ = 103, $n_{IRBL20-IR24}$ = 129, $n_{IRBLi-F5}$ = 108, $n_{IRBL3-CP4}$ = 94, $n_{IRBLk-K}$ = 144, $n_{IRBLsh-S}$ = 126, $n_{IRBLKs-S}$ = 123, $n_{IRBL9-W}$ = 53). LTH Lijiangxintuanheigu. The relative fluorescence intensity is color-coded: red, low; green, medium; and blue, high fluorescence. PM plasma membrane, FM4 = FM4-64. Data are means ± SE; \*\**P* < 0.01 (independent-samples two-sided Student's *t* test in images **l** and **x**). Scale bar = 1 cm in images **a–k**; Scale bar = 10 μm in images **m–w**.

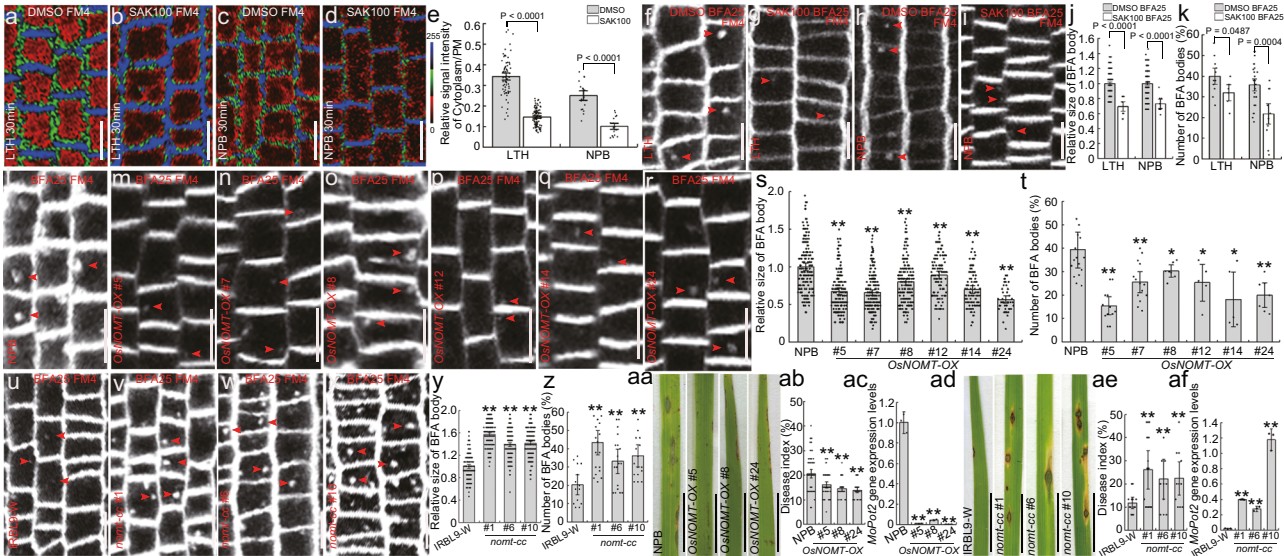

**Fig. 2 | Sakuranetin attenuates endocytosis in rice root cells and confers resistance against rice blast in rice lines overexpressing *OsNOMT*.** The root epidermal cells of rice seedlings LTH (**a**, **b**) and NPB (**c**, **d**) were treated for 90 min with 100 μM SAK (**b**, **d**) or with an equivalent volume of DMSO as a control (**a**, **c**). **e** Quantification of fluorescence intensity in the plasma membrane versus the cytoplasm shown in images **a–d** ($n_{LTH/DMSO}$ = 69; $n_{NPB/DMSO}$ = 19; $n_{LTH/SAK}$ = 100; $n_{NPB/SAK}$ = 18). The root epidermal cells of rice lines LTH (**f**, **g**) and Nipponbare (**h**, **i**) were treated for 90 min with either 25 μM BFA, 100 μM SAK and labeled with 4 μM FM4-64 (**g** and **i**), or with 25 μM BFA, 4 μM FM4-64 labeling and an equivalent volume of DMSO as a control (**f**, **h**). **j** Quantification of the relative size of the BFA bodies shown in images **f–i** ($n_{LTH/DMSO}$ = 125, $n_{NPB/DMSO}$ = 177, $n_{LTH/SAK}$ = 10, $n_{NPB/SAK}$ = 8). **k** Quantification of the number of BFA bodies in cells shown in images **f–i** ($n_{LTH/DMSO}$ = 11, $n_{NPB/DMSO}$ = 26, $n_{LTH/SAK}$ = 6, $n_{NPB/SAK}$ = 12). Root epidermal cells of rice wild-type Nipponbare (**l**) and transgenic lines overexpressing *OsNOMT* (**m–r**) were treated with 25 μM BFA and 4 μM FM4-64 labeling for 90 min. **s** Quantification of the relative size of the BFA bodies shown in images **l–r** ($n_{NPB}$ = 185; $n_{#5}$ = 176; $n_{#7}$ = 213; $n_{#8}$ = 173; $n_{#12}$ = 110; $n_{#14}$ = 97; $n_{#24}$ = 47). **t** Quantification of the number of BFA bodies in cells shown in images **l-r** ($n_{NPB}$ = 12; $n_{#5}$ = 12; $n_{#7}$ = 14; $n_{#8}$ = 7; $n_{#12}$ = 5; $n_{#14}$ = 5; $n_{#24}$ = 7). The root epidermal cells of the wild-type rice IRBL9-W (**u**) and the rice *nomt-cc* mutant lines (**v–x**) were treated with 25 μM BFA and 4 μM FM4-64 labeling for 90 min. **y** Quantification of the relative size of BFA bodies shown in

images **u–x** ($n_{IRBL9-W}$ = 162; $n_{#1}$ = 241; $n_{#6}$ = 186; $n_{#10}$ = 244). **z** Quantification of the number of BFA bodies in cells shown in images **u–x** ($n_{IRBL9-W}$ = 14; $n_{#1}$ = 20; $n_{#6}$ = 18; $n_{#10}$ = 17). Leaf phenotypes of rice wild-type Nipponbare, the *OsNOMT*-over-expressing rice lines #5, #8 and #24 (**aa**), rice IRBL9-W and *nomt-cc* mutant lines #1, #6 and #10 (**ad**). Seedlings were inoculated with spores of the fungus *M. oryzae*, strain Guy11 and grown for 7 days. **ab**, **ae** Quantification of disease index of the rice lines shown in image **aa** ($n_{NPB}$ = 72; $n_{#5}$ = 58; $n_{#8}$ = 47; $n_{#24}$ = 38) and image **ad** ($n_{IRBL9-W}$ = 13; $n_{#1}$ = 14; $n_{#6}$ = 11; $n_{#10}$ = 11). The expression levels of the *MoPot2* gene were assessed in the leaves of Nipponbare and the lines overexpressing *OsNOMT* (n = 3 biologically independent experiments) (**ac**), rice IRBL9-W and the *nomt-cc* mutant lines (n = 3 biologically independent experiments) (**af**). The relative fluor-escence intensity is color-coded: red, low; green, medium; and blue, high fluores-cence. NPB = Nipponbare, LTH = Lijiangxintuanheigu, *OsNOMT-OX* = Rice Nipponbare expressing *3SS::OsNOMT*, PM plasma membrane, SAK sakuranetin. Data are means ± SE; \**P* < 0.05, \*\**P* < 0.01. *P* values were generated with independent-samples two-sided Student's *t*-test in images **e**, **j**, **k**, **s**, **t**, **y**, **z**, **ab** and **ae** and independent-samples two-sided SPSS analysis in images ac and af. Scale bar = 10 μm in images **a–d**, **f–i**, **l–r**, and **u–x**. Scale bar = 1 cm in images **aa** and **ad**. The red arrowheads indicate the BFA bodies.

treatment could modify membrane rigidity and slow down the for-mation of endocytic vesicles, we assessed the recoveries of fluores-cence intensity by using a fluorescence recovery after photobleaching (FRAP) method in rice root PM treated with exogenous and higher or lower endogenous sakuranetin. The fluorescence intensity of the root epidermal cells of the wild-type rice lines LTH (Supplementary Fig. 13b, c) and NPB (Supplementary Fig. 13e, f) stained with FM4-64 and treated with sakuranetin did not show significant differences compared to the mock-treated controls (Supplementary Fig. 13a, d). To test whether the effect of sakuranetin on the fluorescence recovery involves de novo protein synthesis, the roots of wild-type LTH and NPB

seedlings were treated with sakuranetin in the presence of the protein synthesis inhibitor cycloheximide (CHX). We found that sakuranetin did not impair the fluorescence recovery of root PM in LTH (Supple-mentary Fig. 13g–i) and NPB even in the presence of CHX (Supple-mentary Fig. 13j–l).

We then detected fluorescence intensity in the roots of rice seedlings with higher or lower endogenous sakuranetin content, and found that the fluorescence recovery in the roots of rice seedlings overexpressing *OsNOMT* (#5, #8 and #24) (Supplementary Fig. 13n–q) and the *nomt*-cc mutant lines (#1, #6 and #10) (Supplementary Fig. 13s–v) did not show significant differences compared to their

controls (Supplementary Fig. 13m, r). These results indicate that the sakuranetin treatment does not affect PM fluidity.

## Sakuranetin attenuates CME of effector of the fungus *M. oryzae* and conveys resistance to rice blast

Sakuranetin acts as a phytoalexin and is highly expressed in rice plants infected by the fungus *M. oryzae*[46]. To gain additional insight into a potential role of sakuranetin attenuation of endocytosis in resistance against rice blast, the lines #5, #8 and #24 overexpressing *OsNOMT* and wild-type NPB rice seedlings were inoculated with spores of the fungus *M. oryzae* Guy11 (Fig. 2aa). The transgenic rice lines showed higher resistance to rice blast (Fig. 2ab), accompanied with decreased levels of *MoPot2*, which is a transposon used to detect the genetic diversity of *M. oryzae*[47,48], compared with the wild-type NPB plants (Fig. 2ac). Conversely, the *nomt-cc* mutant lines #1, #6 and #10 (Fig. 2ad) were susceptible to rice blast (Fig. 2ae) and had increased *MoPot2* levels (Fig. 2af) compared with control IRBL9-W seedlings (Fig. 2ad) when infected with *M. oryzae*. Indeed, when the wild-type rice lines LTH (Supplementary Fig. 14a) and NPB (Supplementary Fig. 14c) were treated with different concentrations of sakuranetin (25 µM, 50 µM and 100 µM), the resistance to rice blast conferred correlated with the increases in sakuranetin levels (Supplementary Fig. 14b, d), similar to the observed attenuations in endocytosis following treatment with different concentrations of sakuranetin (Supplementary Fig. 6). Together, these data demonstrate that sakuranetin is able to confer resistance against rice blast.

The PWL2 effector from the fungus *M. oryzae* can be translocated into the rice cytoplasm and can even move between rice cells[14]. To check whether sakuranetin was able to attenuate trafficking of this effector, we constructed two rice lines LTH/*35S::PWL2-GFP* and IRBL9-W/*35S::PWL2-GFP*. The IRBL9-W/*35S::PWL2-GFP* line also has higher sakuranetin levels in the roots (Supplementary Fig. 15). We found that the PWL2-GFP bound to the membrane in all cases (Fig. 3a–f). To visualize PWL2-GFP trafficking, we treated the transgenic *35S::PWL2-GFP* rice lines (LTH and IRBL9-W) with BFA and found that the size of the PWL2-GFP BFA compartments was smaller in the IRBL9-W/*35S::PWL2-GFP* seedlings (Fig. 3b, d, f, g) than in the LTH/*35S::PWL2-GFP* seedlings (Fig. 3a, c, e, g). When the rice line LTH/*35S::PWL2-GFP* was co-treated with sakuranetin and BFA (Fig. 3i), the sakuranetin significantly decreased the formation of the PWL2-GFP compartment as compared to the control treated only with BFA (Fig. 3h, j).

To further investigate whether sakuranetin decreases secretion of the *M. oryzae* effector into the rice cell cytoplasm, the rice line LTH was inoculated with *M. oryzae* Guy11 expressing the *M. oryzae* avirulence protein AVR1-CO39[49,50] fused with red fluorescent protein (*AvrCO39-mRFP*). Compared with the control treated only with DMSO, the AvrCO39-mRFP fluorescence intensity at the plasma membrane was significantly decreased following sakuranetin treatment (Supplementary Fig. 16). This all shows that sakuranetin attenuates the internalization and trafficking of the *M. oryzae* effectors, providing a plausible mechanism for its positive effect on rice resistance against rice blast.

Next, we tested which cellular endocytic mechanism is targeted by sakuranetin. To test involvement of the major endocytic mechanism, clathrin-mediated endocytosis (CME)[51], we used transgenic rice lines NPB/*ProOsPIN1b::OsPIN1b-GFP* (Fig. 3k, l) and NPB/*ProOsPIN2::OsPIN2-GFP* (Fig. 3m, n), which express PIN proteins – the established cargo of CME[52]. The transgenic lines were co-treated with the CME inhibitor TyrA23, BFA and sakuranetin. The similar size of the BFA bodies in *ProOsPIN1b::OsPIN1b-GFP* (Fig. 3k, l) and *ProOsPIN2::OsPIN2-GFP* (Fig. 3m, n) treated with or without sakuranetin indicated that TyrA23 treatment rendered roots insensitive to sakuranetin in terms of endocytosis inhibition (Fig. 3k–o). Furthermore, we tested the rice line LTH/*35S::PWL2-GFP*. TyrA23 treatment decreased the size of the PWL2-GFP compartments (Fig. 3q, s) compared with the control treated with

BFA alone (Fig. 3p), demonstrating that PWL2 is also internalized by CME. When the LTH/*35S::PWL2-GFP* line was co-treated with sakuranetin, TyrA23 and BFA (Fig. 3r), there was no significant difference in PWL2-GFP compartment formation compared with the control co-treated with TyrA23 and BFA only (Fig. 3q, s) suggesting that, similar to that found for PINs, sakuranetin cannot affect endocytic trafficking of PWL2 when CME is inhibited. These data show that sakuranetin decreases CME of both PIN plasma membrane proteins and the fungal effector PWL2.

## Rice lines showing attenuated CME have enhanced resistance to rice blast

Since attenuation of endocytosis by sakuranetin occurred via CME (Fig. 3k–s), we further investigated the resistance of *chc-cc* lines #1 and #2 (Fig. 4a). These *chc-cc* mutants have higher resistance to rice blast compared to control NPB plants (Fig. 4a–c). We then treated NPB seedlings with endosidin9 (ES9), an inhibitor of CME[53], prior to the inoculation with *M. oryzae* Guy11 spores (Fig. 4d). ES9-treated seedlings showed clearly enhanced resistance to rice blast as compared to the control (Fig. 4e, f), supporting the observations from the *chc-cc* mutants (Fig. 4a–c). Endosidin9-17 is a more specific version of Endosidin9 lacking cytoplasmic acidification activity, and both ES9 and ES9-17 inhibit clathrin heavy chains[54]. To test whether the effect of ES9 enhanced rice resistance to rice blast is associated with its ability to induce cytoplasmic acidification, wild-type rice NPB seedlings were treated with ES9-17 following with inoculation with *M. oryzae* Guy11. Rice seedlings subjected to ES9-17 treatment showed increased resistance to rice blast compared to the control (Fig. 4g–i). These results show that sakuranetin-decreased CME correlates with and is sufficient to confer resistance to rice blast.

## Discussion

Rice blast induced by *M. oryzae* is the most serious disease of rice worldwide, and sakuranetin is a phytoalexin known to be involved in immunity against it. Recently, it has been found that R genes regulate a number of genes encoding enzymes for secondary metabolites related to plant immunity, including phytoalexins such as sakuranetin[46]. In this study, we showed a mechanism underlying sakuranetin's role in defense against rice blast. We show that resistant rice lines expressing R genes had attenuated endocytic recycling of PM proteins, and that this was associated with increased levels of sakuranetin, which alone are sufficient to decrease the endocytosis of PM proteins. This sakuranetin-mediated attenuation of endocytosis targets the clathrin-mediated pathway.

Resistant rice lines hamper the constitutive endocytosis of PM proteins as well as the endocytic uptake of a fluorescent tracers, correlated with higher endogenous levels of sakuranetin. Furthermore, sakuranetin treatment enhances rice resistance to rice blast and decreases endocytosis of the fungal effector PWL2. These observations consistently suggest that resistant rice lines attenuate endocytosis of fungal effectors by secreting phytoalexins, including sakuranetin, to enhance immunity against rice blast. Moreover, our research also implies that the functions of R genes and their underlying mechanisms, which are as yet not fully understood, might involve the attenuation of endocytosis of fungal effectors as a more general mechanism in plant defense.

The endocytosis of PM proteins is decreased in rice lines over-expressing the sakuranetin biosynthesis enzyme OsNOMT or by exogenous sakuranetin treatment, but is promoted in the *nomt*-cc mutants. Sakuranetin attenuates endocytosis via the CME pathway (Fig. 3k–s). Sakuranetin inhibits the growth of the fungus *M. oryzae*[46], although the mechanism by which sakuranetin hampers the secretion of the effector into rice cells by *M. oryzae* needs further investigation, it is possible that it is also related to the effect of sakuranetin on clathrin function.

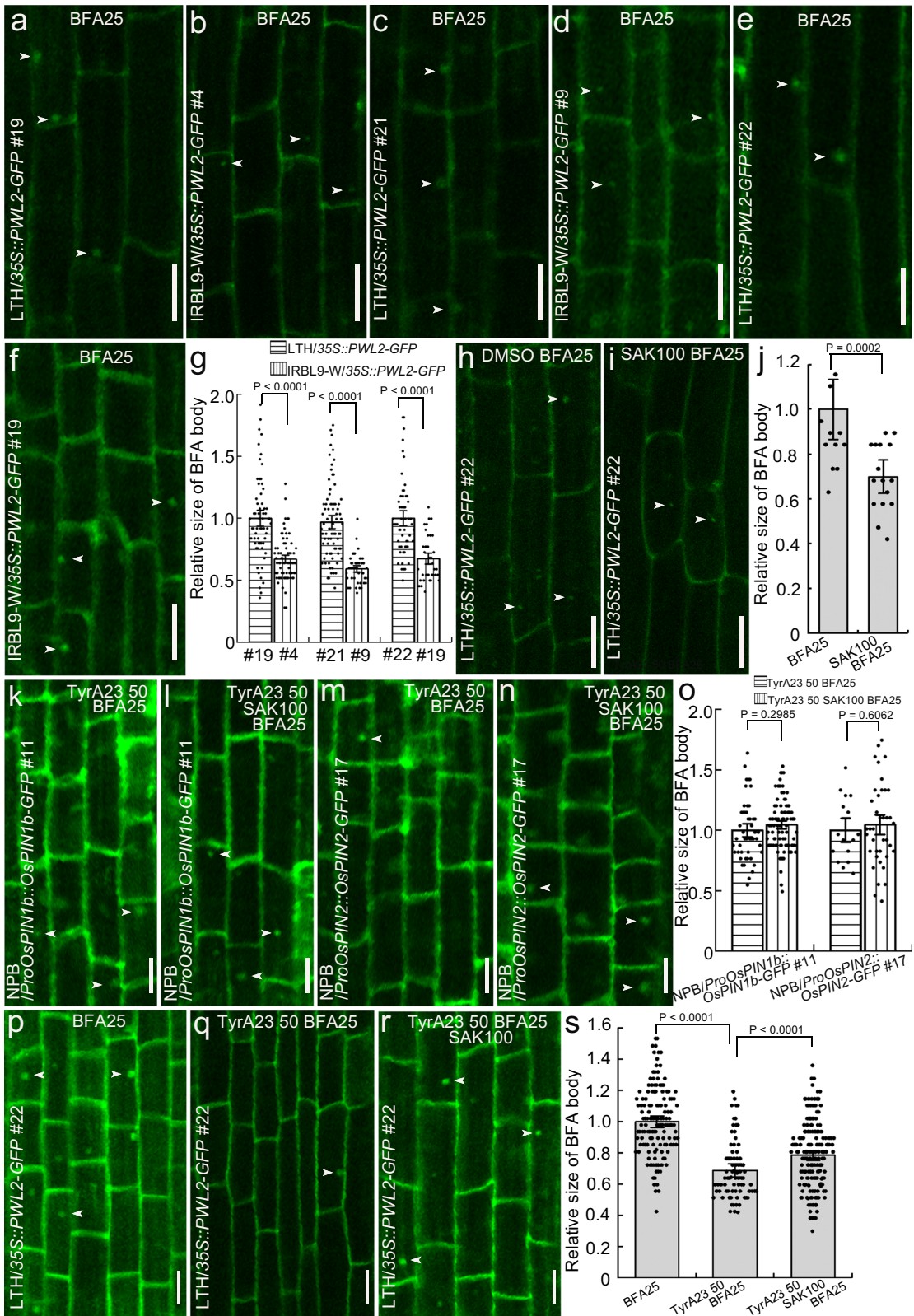

Endocytosis plays a crucial role in plant immunity against different pathogens[55], and sakuranetin is a phytoalexin involved in rice defense[46]. Following exogenous sakuranetin treatment, rice lines overexpressing *OsNOMT* and clathrin mutants show an enhanced immune response against the fungus *M. oryzae*. Sakuranetin mediated attenuation of the endocytosis of *M. oryzae* effector PWL2 is dependent on the clathrin-mediated pathway (Fig. 3). Endocytosis of the pattern recognition receptor (PRR) of MAMP (Microbe-associated molecular pattern) is proposed as an early signaling event after signal perception, reflecting ligand-induced endocytosis[20]. In contrast, the decrease of constitutive endocytosis by sakuranetin can be considered as a late event requiring firstly the phytoalexin synthesis and then its secretion for plant resistance. Overall, sakuranetin down-regulating CME provides a key mechanism, by which rice can defend against rice

**Fig. 3 | Sakuranetin decreases endocytosis of the PWL2 protein via the CME pathway in rice root epidermal cells.** The root epidermal cells of rice lines LTH/*3SS::PWL2-GFP* (**a**, **c**, **e**) and IRBL9-W/*3SS::PWL2-GFP* (**b**, **d**, **f**) treated with 25 µM BFA for 90 min. **g** Quantification of the relative size of the BFA bodies shown in the images **a**–**f** ($n_{LTH/3SS::PWL2-GFP\ \#19} = 63$; $n_{LTH/3SS::PWL2-GFP\ \#21} = 76$; $n_{LTH/3SS::PWL2-GFP\ \#22} = 56$; $n_{IRBL9-W/3SS::PWL2-GFP\ \#4} = 71$; $n_{IRBL9-W/3SS::PWL2-GFP\ \#9} = 44$; $n_{IRBL9-W/3SS::PWL2-GFP\ \#19} = 38$). The root epidermal cells of seedlings of the rice line LTH/*3SS::PWL2-GFP* were co-treated for 90 min with 100 µM SAK and 25 µM BFA (**i**), or with 25 µM BFA and an equivalent volume of DMSO as a control (**h**). **j** Quantification of the relative sizes of the BFA bodies shown in the images **h**, **i** ($n_{DMSO} = 15$; $n_{SAK} = 17$). The root epidermal cells of rice NPB/*ProOsPIN1b::OsPIN1b-GFP* (#11) (**k**, **l**) and NPB/*ProOsPIN2::OsPIN2-GFP* (#17) (**m**, **n**) co-treated for 90 min with either 100 µM SAK, 25 µM BFA and 50 µM TyrA23 (**l**, **n**), or with 25 µM BFA, 50 µM TyrA23 and an equivalent volume of DMSO as a control (**k**, **m**). **o** Quantification of the relative size of the BFA bodies shown in the images **k**–**n** (DMSO: $n_{NPB/ProOsPIN1b::OsPIN1b-GFP} = 48$; $n_{NPB/ProOsPIN2::OsPIN2-GFP} = 16$; SAK: $n_{NPB/ProOsPIN1b::OsPIN1b-GFP} = 92$; $n_{NPB/ProOsPIN2::OsPIN2-GFP} = 39$). The root epidermal cells of rice LTH/*3SS::PWL2-GFP* were co-treated for 90 min with 25 µM BFA and DMSO as a control (**p**), or with 50 µM TyrA23 and 25 µM BFA (**q**), or with 100 µM SAK, 50 µM TyrA23 and 25 µM BFA (**r**). **s** Quantification of the relative sizes of the BFA bodies shown in the images **p**–**r** (n = 143 in image **p**; n = 80 in image **q**; n = 188 in image **r**). SAK sakuranetin, TyrA23 tyrphostin A23. Data are means ± SE; *P* values were generated with independent-samples two-sided Student's *t*-test in images **g**, **j**, **o** and **s**. Scale bar = 20 µm in images **a**–**f**, **h** and **i**, Scale bar = 10 µm in images **k**–**n** and **p**–**r**. Arrowheads indicate the BFA bodies.

blast and, possibly, also other plant species against diverse pathogen infections. Identification of the mechanism underlying the effect of sakuranetin on CME and elucidation of the relevant regulatory mechanisms affected by sakuranetin remain challenging topics for future investigations.

## Methods

### Plant materials and fungus growth conditions

The rice lines used in this study were *3SS::OsNOMT* (in a NPB background), *chc-cc* (in a NPB background), *ProOsPIN1b::OsPIN1b-GFP* (in a NPB background), and *ProOsPIN2::OsPIN2-GFP* (in a NPB background), LTH, rice NILs (in a LTH background), *nomt-cc* (in a IRBL9-W background), *3SS::PWL2-GFP* (expressing in a LTH and IRBL9-W background, respectively), *3SS::OsPIN3t-GFP* (in a Zhonghua 11 background)[56], resistance rice line KY131-*Pigm* (in a KY131 background)[11], *NahG*[34] (in a NPB background), *ipa1*[33] (in a Zhonghua 11 background), *3SS::miR393a*[38] (in a Zhonghua 11 background), *arf12*[36] (in a Dong Jin background), *OsNPR1*-RNAi[35] (in a TP309 background), and *OsROD1*-overexpression[37] (in a TP309 background). The seeds of rice varieties LTH, rice NILs, the overexpression and mutant lines, and resistance line KY131-*Pigm* were grown on 1/2 MS medium in vertically oriented plates at 28 °C with 12 h light. The *M. oryzae* Guy11 expressing *AvrCo39-mRFP* (in a Guy11 background) was cultured at 28 °C on potato dextrose agar medium (potato 200 g L$^{-1}$, glucose 10 g L$^{-1}$, agar 18 g L$^{-1}$).

### Inoculation of rice seedlings with *M. oryzae*

Seedlings of rice Nipponbare (NPB), LTH, IRBL9-W, *35 S::OsNOMT*, *nomt-cc*, *chc-cc* and NILs were grown in a greenhouse at 28 °C. After the seedlings had developed three leaves, they were inoculated with a spore suspension of *M. oryzae* strain Guy11 at $1 × 10^4$ conidia mL$^{-1}$, containing 0.02% Tween 20. The inoculated seedlings were grown in growth chambers at 28 °C for 3 days, and then treated with 50 µM Endosidin 9 (SML2706; Sigma-Aldrich Trading Co., Ltd, Shanghai, China) dissolved in water. Seedlings treated with sterilized water acted as controls. To analyze the disease index, the number of leaves with the same lesion level was multiplied by the corresponding lesion level. The disease severity values were then added to obtain the final disease severity value across all the leaves. The disease index was obtained by dividing the final disease severity value by the total number of leaves and the highest lesion level. This method of disease index analysis follows that in published reports[57,58]. For punch inoculation, the leaves of two-month old seedlings grown in the field were punctured and inoculated with mycelial discs for 3 days, and then treated either with 25 µM, 50 µM or 100 µM sakuranetin (DY0194, Chengdu DeSiTe Biological Technology company, Chengdu, China), or with 50 µM Endosidin 9-17 (HY-131683, Med-ChenExpress, Shanghai, China) dissolved in water, and leaves treated with sterilized water acted as a control. The lesion lengths were measured after 7 days of leaves inoculated with mycelial discs using Image J 1.41 software.

### Inoculation of rice leaf sheaths with the fungus *M. oryzae*

Leaf sheaths of rice seedlings with 4-5 developed leaves were injected using a syringe with a spore suspension of *M. oryzae* strain Guy11 (*AvrCo39-mRFP*) at $1 × 10^4$ conidia mL$^{-1}$ containing 0.02% Tween 20, and were grown at 28 °C for 5 days. The sides, epidermal layer and mid-vein cells of the leaf sheaths were removed, leaving sections three to four cell layers thick. These sections were then treated for 120 min with either 100 µM sakuranetin or an equivalent volume of DMSO as a control. The specimens were then examined under a Leica SP5 confocal microscope. The relative fluorescence intensity of the cell membrane of the rice leaf sheath was measured with Image J 1.41 software. In order to quantify fluorescence intensity without the risk of background noise, we measured the fluorescence intensity of the cell membrane of the rice leaf sheath and the area without the leaf sheath. We obtained the true fluorescence value by subtracting the one from the other.

### Observation of endocytosis of rice root cells

We observed endocytosis in rice leaf epidermal cells and found that the endocytic marker FM4-64 was not able to effectively stain leaf epidermal cells, and rice root epidermal cells labeled with FM4-64 were therefore used to observe cellular endocytosis. The roots of 5-day old rice seedlings were then pre-treated with 100 µM sakuranetin for 30 min, and then treated with 100 µM sakuranetin for 90 min with a solution containing sterilized water together with the following chemicals at the final concentration indicated: 4 µM FM4-64 (stock in water; Thermo Fisher Scientific Inc, Waltham, United States), 50 µM TyrA23 (MedChenExpress, Shanghai, China) and 25 µM BFA (Med-ChenExpress, Shanghai, China) DMSO-prepared stock solutions. Where rice seedlings were treated with FM4-64 alone, the root tips were incubated with 4 µM FM4-64 for 90 min on ice, and then the root tips used to observe the fluorescence intensity for 20 or 30 min. Confocal images were obtained with a Leica SP5 confocal microscope equipped with a ×40 objective, the fluorescence of FM4-64 was excited at 488 nm and emission was detected between 640 nm and 691 nm. The fluorescence intensities of the cell membrane and cytoplasm were measured using the Image J 1.41 software. To measure the fluorescence intensity of the cell membrane, 'segmented line' with size 2 was selected, 'measure' in 'Analyze' was clicked and the mean value was recorded. To measure the fluorescence intensity of the cytoplasm, 'Polygon selections' was selected, 'measure' in 'Analyze' was clicked and the mean value was recorded. The ratio of fluorescence intensity of the cell membrane to that of the cytoplasm was calculated from the mean value of the fluorescence intensity in the cell cytoplasm and that in the cell membrane, and was used to indicate the effect of sakuranetin on endocytosis.

### Rice root subcellular structure examined with electron microscopy

Roots of five-day old rice seedlings were treated for 90 min with 25 µM BFA and either 100 µM sakuranetin or an equivalent volume of DMSO

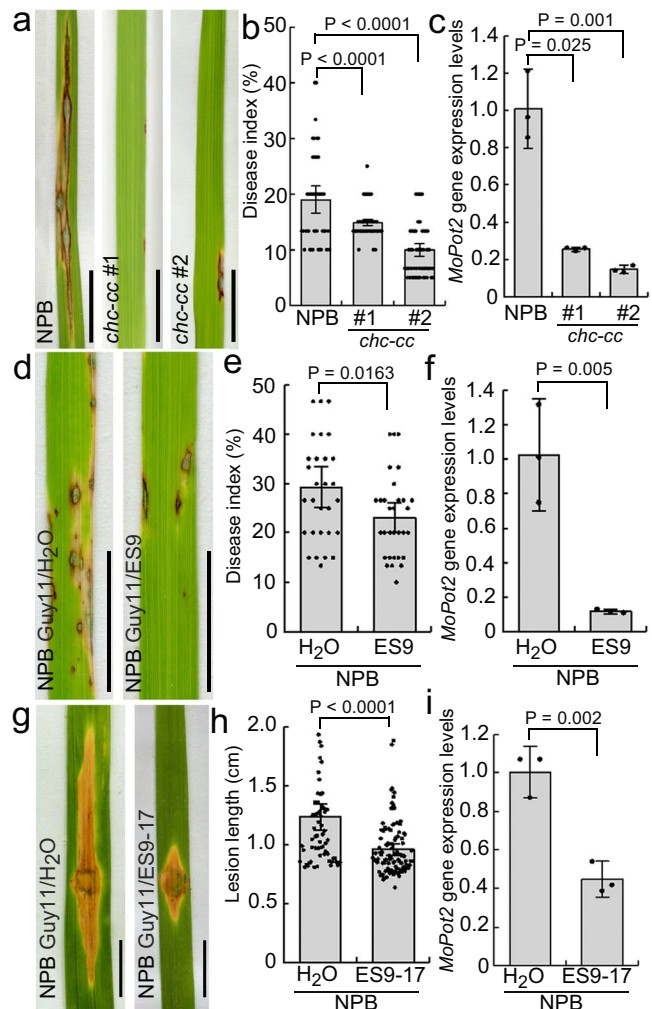

**Fig. 4 | Resistance to rice blast in the rice *chc-cc* mutant and Nipponbare lines following treatment with ES9 and ES9-17.** Leaf phenotypes of rice wild-type Nipponbare (**a**, **d**, **g**) and *chc-cc* mutant lines #1 and #2 (**a**). Seedlings were inoculated with spores of the fungus *M. oryzae*, strain Guy11, and grown for 3 days. The wild-type Nipponbare was also treated with either 50 µM endosidin9 (**d**) or 50 µM endosidin9-17 (**g**), or was treated with sterilized water as a control (**d**, **g**). **b**, **e** Quantification of disease index of the rice shown in image a ($n_{NPB}$ = 46; $n_{#1}$ = 79; $n_{#2}$ = 78) and image d ($n_{DMSO}$ = 31; $n_{ES9}$ = 31). **h** Quantification of lesion length of the rice shown in image g ($n_{DMSO}$ = 64; $n_{ES9-17}$ = 102). **c**, **f**, **i** The expression levels of the *MoPot2* gene were assessed (n = 3 biologically independent experiments) in the leaves shown in the images **a**, **d**, and **g**. The actin gene was used as the control gene to check the levels of *MoPot2* using real-time PCR. NPB Nipponbare, ES9 endosidin9, ES9-17 endosidin9-17. Data are means ± SE; *P* values were generated using independent-samples two-sided Student's *t* test in images **b**, **e** and **h** and independent-samples two-sided SPSS analysis in images **c**, **f** and **i**. Scale bar = 1 cm.

(as a control). For the examination of the subcellular structure, we used transmission electron microscopy (TEM), and followed the standard protocol for sample preparation. The meristem region of a primary root was cut into small pieces (1 × 1 mm), fixed with glu-taric dialde-hyde and osmium (VIII) oxide twice, and, after dehydration with ethyl alcohol, the pieces were embedded in a resin containing the agents Dow epoxy resin (DER) 736, 1-nonenylsuccinic anhydride (NSA), ali-phatic epoxy resin ERL-4221 and dimethylaminoethanol (DMAE). The samples were sectioned to a thickness of 70 nm in a LEICA EM UC7 ultramicrotome. They were then stained with uranyl acetate (Beijing Zhongjingkeyi Technology Co., Ltd, Beijing, China) for 15 min and with alkaline lead citrate, containing trisodium citrate dehydrates, lead nitrate, and sodium hydroxide, for 5 min. The specimens were then examined using a FEI TECNAI SPIRIT G2 TEM under a voltage of 80 kV and a current of 27 A.

## Measurement of sakuranetin
To measure levels of sakuranetin, rice seedlings were cultured on 1/2 MS medium for 2 days in darkness, and were then grown in vertically oriented plates at 28 °C with 12 h light for a further 5 days. Leaf and root samples were harvested and ground into powder in liquid nitrogen. Each sample (~200 mg) was mixed with 1 mL of pre-cooled ethyl acetate supplemented with 20 ng of $D_6$-JA (HPC Standards GmbH, Germany) as the internal standards. After vortexing for 10 min, the samples were centrifuged at 14,000 × *g* for 15 min. The supernatants were carefully transferred to new 2 mL microfuge tubes and then completely dried at 45 °C in a vacuum concentrator (Eppendorf). Each of the dried pellets was suspended in 0.2 mL of 50% methanol by vigorous vortexing for 10 min. After centrifugation at 14,000 × *g* for 15 min at 4 °C, 0.1 mL of each supernatant was transferred to a glass vial with inserts, and samples were analyzed on an HPLC-MS/MS system (LCMS-8040 system, Shimadzu) installed with a Shim-pack XR-ODS III column (2.0 mm I.D. × 75 mm L., 1.6 µm, Shim-pack). A gradient mode was employed as follows: mobile phase consisted of solvent A (0.05% formic acid and 5 mM ammo-nium formate in water) and solvent B (methanol). Gradient elution was carried out at 10–30% B over 3 min, 30–95% B over 4.5 min, kept at 95% B for 3 min, and then re-equilibrated to 90% A for 2 min and kept at 90% A for a further 3.5 min. A negative electrospray ioniza-tion mode was used for detection. For sakuranetin, a specific parent ion with a unique mass-to-charge (m/z) ratio (285.00) was selected and fragmented, generating a daughter ion (119.00). The chroma-togram for each compound was obtained based on the specific daughter ions. The peak areas of $D_6$-JA and sakuranetin were mea-sured, and the concentration of each sakuranetin was determined using the standard curve of sakuranetin normalized with the area of $D_6$-JA.

## RNA isolation and reverse transcription reaction
Total RNA from rice leaves and roots was isolated using an EasyPure® Plant RNA Kit (TransGen Biotech Co., Ltd, Beijing, China). The first strand cDNA was synthesized with DNase I-treated RNA, oligo-dT pri-mer and TransScript All-in-One First-Strand cDNA Synthesis SuperMix for qPCR Kit (TransGen Biotech Co., Ltd, Beijing, China).

## Real-time PCR analysis
The relative quantitative expression levels of the *OsNOMT* (LOC_Os12g13800) and *MoPot2* (Z33638.1) genes were determined using an ABI QuantStudio 7 Flex Real-Time PCR System (Thermo Fisher Scientific lnc, Waltham, United States). The 10 µL PCR mixture was prepared with PowerUp ™ SYBR ™ Green Master Mix (Thermo Fisher Scientific lnc, Waltham, United States) containing the primer pairs OsNOMT-rFP (5′- AAGCGTTCCGGCTCGACGTCAT -3′) and OsNOMT-rRP (5′- ACTCGAGAGCCCAGACGTTGGT -3′), MoPot2-rFP (5′- ACGACCCGTCTTTACTTATTTGG -3′) and MoPot2-rRP (5′- AAG-TAGCGTTGGTTTTGTTGGAT -3′) to amplify *OsNOMT* and *MoPot2*, respectively, as well as a cDNA or genomic DNA template. The actin gene (Os11g0163100) acted as the internal control and was amplified with the primer pair OsActin-FP (5′- GAGTATGATGAGTCGGGTCCAG -3′) and OsActin-RP (5′- ACACCAACAATCCCAAACAGAG -3′). PCR was performed under denaturation at 95 °C for 2 min, followed by 40 cycles of 95 °C for 45 seconds, 54-62 °C for 30 seconds and 72 °C for 1 min. Three biological replicates were made. The relative expres-sion levels were calculated using the $2^{-\Delta\Delta Ct}$ method. The software SPSS Version 19.0 (IBM, Inc., Armonk, NY, USA) was used to analyze differ-ences in gene expression. *P* < 0.05 indicates a statistical difference, and *P* < 0.01 indicates a statistically significant difference.

## Construction of expression vector, CRISPR/Cas9 plasmids, *OsIPA1*, and *OsNOMT* knockout mutants, and rice transformation

The expression vector pATs containing the *PWL2:GFP* fusion gene (*PWL2* accession numbers: U26313.1) and the CRISPR/Cas9 vector bgk032 targeted to *OsNOMT* were constructed at the Biogle Company (Hangzhou Biogle Co., Ltd, Hangzhou, China). The rice mutant lines *nomt-cc* (with an IRBL9-W background) and *ipa1* (with a Zhonghua 11 background) were produced using the CRISPR/Cas9 method at Biogle Company (Hangzhou Biogle Co., Ltd, Hangzhou, China). The specific guide-RNA (gRNA) sequences for *OsNOMT* and *OsIPA1* were designed as follows: 5'-TCTGCACTTGTGGGGGGACG-3' and 5'-GTATGTGCTC CCGGCGAGCC-3' for the *OsNOMT* gene and 5'-GCCGTCCTCGTC TTCCAAGG -3' for the *OsIPA1* gene. The target gene sequences including the protospacer adjacent motif (PAM) of *OsNOMT* were 5'-TCTGCACTTGTGGGGGGACGAGG-3' and 5'-GTATGTGCTCCCGGC-GAGCCCGG-3'for the *OsNOMT* gene and 5'-GCCGTCCTCGTCTTCC AAGGCGG-3' for the *OsIPA1* gene, and the trinucleotides AGG and CGG for the *OsNOMT* gene and CGG for the *OsIPA1* gene acted as the PAMs of *nomt-cc* and *ipa1*, respectively. The CRISPR/Cas9 plasmids were introduced into *Agrobacterium tumefaciens* strain EHA105. Genomic DNA was extracted from rice transformants, and the primer pairs nomt-ccFP (5'-AGGCACAACTACCACTGATGG -3') and nomt-ccRP (5'-GTCACTTTCATGCATCCGGC -3'), and ipa1-FP (5'-CCCAAGCTTGCTG-CACGGTCTCAAGT -3') and ipa1-RP (5'- CGCGGATCCTTGACAGTG CATCTAATGTG -3'), flanking the designed target site were used for PCR amplification. PCR was performed under denaturation at 98 °C for 3 min, followed by 30 cycles of 98 °C for 10 s, 54–58 °C for 20 s and 72 °C for 1 min. The PCR products were sequenced by the Tsingke Company (Tsingke Biogle Co., Ltd, Beijing, China). To obtain trans-genic LTH and IRBL9-W lines containing the *PWL2:GFP* fusion gene, the constructs containing *p35S::PWL2:GFP* were transformed into *Agro-bacterium tumefaciens* strain EHA105. Rice transformation was per-formed using the *Agrobacterium*-mediated method by the Biogle Company (Hangzhou Biogle Co., Ltd, Hangzhou, China). To obtain the expression vector *pBWA(V)HII* containing the fusion genes *OsSYP121-GFP*, *OsARA7-GFP*, or *OsHDEL-GFP*, the constructs containing *ProOs-SYP121:: OsSYP121:GFP*, *ProOsARA7:: OsARA7:GFP* or *ProOsHDEL:: OsH-DEL:GFP*, respectively, were transformed into *pBWA(V)HII* at the Biorun Company (Wuhan BIORUN Bio-Tech Co., LTD, Hangzhou, China).

## Fluorescence recovery after photobleaching (FRAP) analysis

To detect the effect of sakuranetin on cell membrane fluidity, the root tips of 5-day old rice NPB and LTH seedlings were inoculated with 50 μM cycloheximide for 30 min, and then co-treated with 50 μM cycloheximide, 4 μM FM4-64 and 100 μM sakuranetin (dissolved with DMSO) for 90 min, and the root tips pretreated with equivalent volume of DMSO for 30 min and treated with 4 μM FM4-64; or 4 μM FM4-64 and 100 μM sakuranetin; or 4 μM FM4-64, 50 μM cyclohex-imide and the equivalent volume of DMSO for 90 min as a control. Roots of rice overexpressing-*OsNOMT* lines and *nomt-cc* mutant lines were treated with 4 μM FM4-64 for 90 min. Confocal images of root tips were obtained with a Leica SP5 confocal microscope equipped with a ×40 objective. Five iterations of a 488 nm line from a 200 mW argon laser operating at 100% laser power were used to photobleach the selected area of the plasma membrane. The recovery of fluores-cence intensity from thirty iterations was recorded, and the values of fluorescence intensity recovery were measured using the confocal microscope's own software.

## Transient transformation and observation of rice protoplasts

Rice protoplasts were prepared from the stems of rice NPB seedlings cultured on 1/2 MS medium for 14 days in darkness. The stems of at least sixty rice seedlings were cut into 0.5 mm strips, transferred to enzyme solution (0.6 M D-mannitol, 10 mM 2-morpholinoethanesulfonic acid

(MES), 1.5% cellulase R10, 0.75% macerozyme R-10, 0.1% BSA, 1 M CaCl₂ and 5 mM β-mercaptoethanol, pH = 5.7), and shaken at 80 rpm for 4 h in darkness. Then the dissolved strip suspensions were filtered through a mesh screen of hole size 40 μm, and the filtered suspension was immediately added to 30 mL W5 solution (154 mM NaCl, 125 mM CaCl₂, 5 mM KCl and 2 mM MES, pH = 5.7) and shaken by hand for 20 seconds. The protoplasts were centrifuged at 350 g for 4 min at 4 °C, after which 2 mL MMG solution (0.2 M D-mannitol, 0.1 M CaCl₂ and 0.8 g PEG4000, pH = 5.7) was added and the samples were left for 20 min on ice. The number of protoplasts was then adjusted to $1 \times 10^6$ conidia mL⁻¹. To obtain protoplast cells expressing the *OsSYP121-GFP*, *OsARA7-GFP*, or *OsHDEL-GFP* fusion genes, protoplasts cells were transformed using 5 μg of plasmid DNA and incubated at 28 °C for 16 h in darkness. The protoplast cells expressing *OsSYP121-GFP* were pretreated with 100 μM sakuranetin for 15 min, and then co-treated with the following chemicals at the final concentration indicated: 4 μM FM4-64, 100 μM sakuranetin and 25 μM BFA for 30 min used to observe endosome aggregation. The protoplast cells expressing *OsSYP121-GFP*, *ARA7-GFP* and *HDEL-GFP*, respectively, were pretreated with 100 μM sakuranetin for 15 min, and then co-treated with the 4 μM FM4-64 and 100 μM sakuranetin for 30 min on ice used to observe endosomal motility. Protoplast cells expressing *OsSYP121-GFP* treated with 100 μM sakuranetin were incu-bated on ice for 45 min and were used to observe cellular endocytosis. The protoplast cells were examined using a confocal microscope (Leica Microsystem, Wetzlar, Germany; TCS SP8). The fluorescences of FM4-64 and GFP were excited at 488 nm and 633 nm wavelengths, respec-tively; the FM4-64 fluorescence emission was detected between 640 nm and 691 nm, and GFP fluorescence emission was detected between 495 nm and 565 nm. The fluorescence intensity of the plasma mem-brane and cytoplasm of protoplast cells was measured using the Image J 1.41 software.

## Reporting summary

Further information on research design is available in the Nature Portfolio Reporting Summary linked to this article.

## Data availability

All data supporting the findings of this study are available within the paper, its Supplementary Information file, and the Source Data file. The LC-MS/MS data and microscopy data have been deposited at the public repository figshare with the dataset identifier SeRrbJ609 [https://doi.org/10.6084/m9.figshare.25448191.v3]. Original data points in graphs and qPCR data are shown in the Source Data files. Source data are provided with this paper.

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

## Acknowledgements

We thank Professor Jianqiang Wu (Kunming Institute of Botany, Chinese Academy of Sciences) for his generous support with the sakuranetin measurement. We thank International Rice Research Institute (IRRI) for provision of the rice NILs. We thank Professor Zhongkai Zhang (Yunnan Academy of Agricultural Sciences) for his generous support with subcellular structure observation of rice roots. We thank Professor Barbara Valent (Kansas State University) for her generously provision of the plasmid containing the PWL2 gene. We thank Professor Zuhua He (Chinese Academy of Sciences) for the gift of the transgenic rice line expressing the *Pigm* gene, *OsNPR1*-RNAi mutant line and *ROD1*-overexpression rice line. We thank Professor Yinong Yang (The Pennsylvania State University) for provision of the rice line *NahG*. We thank Professor Muyuan Zhu (Zhejiang University) for provision of the transgenic plants overexpressing miR393a. We thank Professor Zhengge Zhu (Hebei Normal University) for provision of the rice line overexpressing *OsPIN3t-GFP*. We thank Professor Yanhua Qi (Zhejiang University) for provision of the *arf12* mutant line. Thanks also go to Professor Jean-Benoit Morel (Plant Health Institute of Montpellier) for provision of the fungus *M. oryzae* strain Guy11 (*AvrCo39-mRFP*). This work was supported by grants from the National Natural Science Foundation of China (Grant Nos. 32260085, 31460453, 31660501, 31860064, 31760500 and 31901870), the Major Special Program for Scientific Research, Education Department of Yunnan Province (Grant No. ZD2015005). The project was also sponsored by SRF for ROCS, SEM (Grant No. [2013] 1792), the Key Projects of Applied Basic Research Plan of Yunnan Province (Grant No. 2017FA018, 202301AS070082), the Major Science and Technology Project in Yunnan Province (202102AE090042, 202202AE090036 and 202102AE090017), the Young and Middle-Aged Academic and Technical Leaders Reserve Talent Program in Yunnan Province (202205AC160076), the China Postdoctoral Science Foundation (2019M653849XB) and the National Key Research and Development Program of China (2023YFE0107500).

## Author contributions

Y.L.D. initiated and supervised the project and designed the experiments; L.H.J., X.Y.Z. and Y.T.Z. performed the majority of the experiments and analyzed and prepared the figures; H.Y.Z., Q.J.F., X.Q.L., W.Y.H., X.Y.Y., X.Z., L.X.W., A.Y., X.H., M.D., Z.A.P., J.Y., L.W.G., J.C.W., H.C.H. and Y.X. performed additional experiments. Y.L.D., J.F., L.H.J., X.Y.Z., Y.T.Z., S.S.Z., C.Y.L., X.H.H. and Y.Y.Z. analyzed the data. Y.L.D. and J.F. wrote and revised the paper with input from all authors.

## Competing interests

The authors declare no competing interests.
