## [Peer Review File · Nature Communications]

Phytoalexin sakuranetin attenuates endocytosis and enhances resistance to rice blastReviewer #1 (Remarks to the Author):

The authors report that NILs containing R proteins activated by *M. oryzae* infection show reduced levels of endocytosis, caused by elevated production of the phytoalexin sakuranetin, and that this prevents infection of the fungus. The evidence is preliminary at best and there are many issues with the manuscript in terms of a lack of data that supports the claims of the authors.

Major comments

1. In Figure 1 the authors take a single susceptible rice line and a range of NILs expressing R proteins. They show that the NILs are resistant (as expected and previously published) and that they show relatively higher cytoplasmic fluorescence of FM4-64, which they use as a proxy for constitutively reduced endocytosis. For this correlation to be made, they need to show that a range of susceptible lines show 'reduced endocytosis' and that genetically diverse resistant lines consistently show elevated endocytosis.
2. In Figure 1, if sakuranetin production is induced by R protein activation, why would these NILs show constitutive reduced endocytosis?
3. In Figure 1, what is the expression of NOMT gene expression in these lines (and in more susceptible lines), and what are the sakuranetin levels?
4. Figure 1: I'm not sure what differences the images from m-w actually show?
5. Figure S3 – sakuranetin treatment with 50 μ M provided resistance. Where was the dilution series to determine this concentration for use? How much is it induced anyway in resistant plants? What is the level of induction in LTH during infection? The authors only looked at one resistant line in this Figure – need to show all of the other resistant lines to demonstrate the correlation with increased sakuranetin production and NOMT expression and also include more susceptible lines to demonstrate this correlation holds up.
6. Figure S5 describes overexpression of NOMT to increase sakuranetin levels. Where are the measurements that show the overexpression increases the phytoalexin production?
7. Figure 2 – overexpression of NOMT leads to reduced BFA body size, whereas NOMT KO is larger. What about resistance to *M. oryzae*? This is not included in this Figure but is instead included in Figure 5 at the end.
8. Figure 3 – I fail to see the significance of this experiment. It shows that transformed rice expressing effector PWL2-GFP shows the effector at the PM and treatment with BFA and SAK results in less endocytosis of it (smaller BFA bodies). This is not a proxy for effector translocation from *M. oryzae*.
9. Figure 4. The fact that use of CME inhibitor and clathrin mutation show the same phenotypes as SAK treatment does not mean that SAK targets CME. This is a correlation rather than a demonstration of genetic association.
10. Figure 5. The new bit is the fact that overexpression of NOMT leads to resistance. This should be included in Figure 2. What does NOMT knock out do? The other components – chc-cc mutation and ES9 treatment repeat the recently published results on endocytosis of *M. oryzae* effectors (Oliveira-Garcia et al 2023) and are therefore not novel.

Minor comments

Introduction – when mentioning the endocytosis of *M. oryzae* effectors it is good to refer to the accompanying paper indicating that oomycete effectors also enter via endocytosis. This broadens the potential impact of your observation on sakuranetin.

Please ensure you have help to correct English errors throughout.

Reviewer #2 (Remarks to the Author):

General comment:

This study assessed the role of the phytoalexin sakuranetin inhibits endocytosis to enhance resistance to rice blast disease. Rice tissue treatment with sakuranetin inhibits endocytosis of plasma membrane localized proteins and fungal effectors translocation. These results were comparable to the treatment with Endosin9, a clathrin heavy chain inhibitor. Overexpression of the sakuranetin biosynthesis gene OsNOMT in rice also inhibits endocytosis and enhances disease resistance. The data analysis and statistics presented in this manuscript are appropriate. However, the authors must show individual data points in all graphs, precise P values and improve the quality of graphs and figures. The manuscript is moderately well written, and the outcomes should

serve as references for fungal geneticists and plant pathologists. I support the publication of this manuscript in Nature Communications after proper corrections.

I have some comments that the authors should address.

Specific comments:

The manuscript was missing line numbers, making it harder for the reviewers to evaluate the manuscript. The authors must resubmit the edited manuscript with numbered lines.

Introduction:

"The plant immune response is critical to plant survival." Please improve the sentence. "The plant immune system is essential to plant survival and adaptation to the environment".

"are able to translocate into rice cells with CME". It should be written: " are translocated into rice cell cytoplasm via CME"

Figure 1: Authors must show individual data points in the graphs of this figure and in all graphs of the manuscript. Please present the graphs in solid colors. Use color-blind friendly colors for the figures. Why the disease index is negative? Use the free space to centralize the graphs L and X, and axel labels should be centralized. Graph legends are touching the axes labels. Size bars should be present on all figure panels. Please give the precise P values.

Figure 2: Same comment regarding the quality of the graph. There is a great discrepancy in the font sizes for the axel's labels. Please give the precise P values.

Figure 3: Same comment regarding the quality of the graph. Please give the precise P values.

Figure 4: Same comment regarding the quality of the graph. Please give the precise P values.

Figure 5: Same comment regarding the quality of the graph. Size bars should be present on all figure panels. Please give the precise P values.

Endosidin9-17 is a more specific version of Endosidin9 lacking cytoplasmic acidification activity. Both ES9 and ES9-17 inhibit clathrin heavy chains.

References: Please double check journal abbreviations and perhaps some appeared twice in two reference lists.

Supplemental Figures: Same comments about graph quality, data points, scale bars, and legend resolution.

Figure S6 is too small. Please improve it.

Figure S8: "Bright" Is it bright field?

Reviewer #3 (Remarks to the Author):

The manuscript presented by Jiang and collaborators aims at a better understanding of immune response and resistance of rice to the pathogenic fungus *Magnaporthe oryzae*, the causal agent of rice blast. The work deals with an intriguing subject related to the negative impact of the main rice

phytoalexin, sakuranetin, on endocytosis of rice cells.

The authors used nicely a combination of genetics, pharmacology and cell biology approaches on different rice lines to correlate the disease resistance with sakuranetin-triggered decrease of clathrin-mediated endocytosis (CME).

This is an innovative work but despite a significant set of complementary experiments (even if some of them sound outside the guideline) some conclusions remain elusive.

The authors used different cultivars without clearly explaining why. The analysis and/or the presentation of the figures /quality of images (such blurred fluorescence and poor resolution) will for some of them to be improved.

Here are listed some detailed areas of concern.

Major points

The introduction indicated that endocytosis process is an important cellular event in plant microbe interaction, and its inhibition can attenuate the induction of plant defense. For instance, it is exemplified through M/D/PAMP-activated internalization by CME of membrane PRRs for degradation in the vacuole. This seems controversial compared with the present work, which showed that a defense component, the phytoalexin sakuranetin, has a negative impact on the endocytosis. Neither in the introduction nor in the conclusion do the authors discuss the spatio-temporality of events. Indeed, PRR endocytosis is proposed as an early signalling event after signal perception, reflecting induced-endocytosis. In contrast, the decrease of constitutive endocytosis by sakuranetin can be considered as a late event requiring firstly the phytoalexin synthesis and secretion for plant resistance.

The authors compared a susceptible line with a set of rice lines resistant to rice blast. Surprisingly, the disease was analysed on rice leaves and the endocytosis was carried out on root cells, using the endocytic marker FM4-64. Why the authors are not analysing the endocytosis of leaf cells, especially since sakuranetin level is higher (fig S3)?

The authors quantified the fluorescence of plasma membrane and cytoplasm to calculate a ratio between the two locations, testifying internalization if high. The method of quantification is not detailed and refers to Du et al 2013 where the inverse ratio was used (cytoplasm/PM) and where inverted black and white images were created, which are easier to analyse (notably when images are saturated as here). Quantification of the size of BFA bodies is a good complement of PM/cyt ratio, but no explanation of the characteristics of the bodies (or shape, number..) is given. The unit of the BFA bodies is missing in the ordinate legend.

Another concern is that the correlation between resistance and degree of inhibition of endocytosis is not clear (fig 1). Indeed, resistant lines present various level of disease index but this is not strictly correlated to the level of inhibition of endocytosis (e.g IRBLsh-S and IRBL9-W present a null disease index but the attenuation of endocytosis is not high at the contrary). Unfortunately, no data on sakuranetin concentration between resistant lines is given.

The same statement could be done with overexpressed plants (fig2r) in which OsNOMT expression is not correlated with attenuation of endocytosis (fig S5, see plant #5 with low expression but endocytosis attenuation at the same level than the other overexpressors). Once again, roots have a lower expression of OsNOMT(which may indicate low levels of sakuratenin) than leaves but endocytosis was analysed in root cells.

Cells treated with sakuranetin present a decrease (not an inhibition as written!) of the endocytosis and of the size of BFA bodies. It will be informative to test a dose-response of sakuranetin treatment.

One main concern is that the sakuranetin is a lipophilic compound that has been shown to be incorporated in cell membranes (e.g see da Cruz Ramos Pires et al 2022, doi: 10.1016/j.colsurfb.2022.112546 112546, ISSN 0927-7765). We can therefore question whether this treatment could modify membrane rigidity and slow down the formation of endocytic vesicles.

The ultrastructure images of rice epidermal cells (fig S7) is not really informative (low quality and contrast) and without quantification. A deeper analysis of such images will be attractive. For example, it seems that dense material is observed inside vacuoles of cells treated by sakuranetin. In such images, it will be also possible to scrutinized membranes or organelles (membrane waving,

nascent endocytic structures, Golgi, Trans Golgi network, multivesicular bodies,...).

The analysis of PWL2-GFP fate is very valuable to link decrease of endocytosis and decrease of the effector entry. The authors indicate that the fusion protein is localized in the plasma membrane, but I think it's better to employ "bound to membrane". The images of figure 3 do not present clear membrane resolution but the quantification of the size of BFA bodies seems convincing.

Nevertheless, the figure title (and the conclusion) should be moderate as the amount of sakuranetin was only analysed in untransformed IRBL9-W(FigS3d) but not in the line expressing PWL2-GFP. Moreover, use "attenuates/decreases" instead of "inhibits".

The images of Fig S8 are not informative in that form, the red fluorescence is almost invisible. How quantification was achieved without running the risk of background noise? Same comment for figure S9. By the way the result of figure S9 (where the title should be change as it is not membrane protein but FM4-64 dye!) are difficult to apprehend and confirm only that sakuranetin has a strong impact on fungal cells. This is at the margin of the study.

Finally, the results with the PIN proteins complete the ones with FM4-64 but even if it show the decrease of endocytosis and inform of the nature of endocytic process, I wonder whether this is the best proteins to analyse, as it is not, in my opinion, a protein directly linked to immune response. Even if I may understand that it is a good tool.

Similarly, the work on root growth and gravity drowns out the main message of the paper.

Minor points

Many edits will have to be carried out. For instance:

- Significance of abbreviations are not always at the first citation.
- Caption fig S1 should be "Endocytosis of plasma membrane..." instead of "endocytosis of plasma membrane proteins". Indeed, FM4-64 reflects also the internalization of the lipid part of the membrane
- Fig2r and Fig S2f: write for the ordinate axis "analyse of surface of PM" instead of "surface of PM protein internalization" as it is FM4-64 labelling!
- Write "endocytosis of root cells" (instead of "endocytosis of root")
- It is not a FM4-64 treatment but a FM4-64 labelling (use treatment for BFA or tyrphostin)

In conclusion, despite significant works using many tools and lines and proposing an original hypothesis, I think that further experiments are needed to confirm and clarify the conclusions drawn. Authors need also to focus on certain results and not spread themselves too thin in order to get a clear message with greater impact.

2nd-November-2023

Dear Reviewers,

In the revised manuscript (ID: NCOMMS-23-20536-T), we have carefully addressed all the comments point-by-point. You will find my response to each comment below.

Reviewer #1 (Remarks to the Author):

The authors report that NILs containing R proteins activated by *M. oryzae* infection show reduced levels of endocytosis, caused by elevated production of the phytoalexin sakuranetin, and that this prevents infection of the fungus. The evidence is preliminary at best and there are many issues with the manuscript in terms of a lack of data that supports the claims of the authors.

Answer: We thank Reviewer 1 for her/his effort and time spent to help us to improve this manuscript. These comments are very valuable to us and have allowed us to improve this text. In the revised text, we have provided data from several new experiments.

Major comments

1. In Figure 1 the authors take a single susceptible rice line and a range of NILs expressing R proteins. They show that the NILs are resistant (as expected and previously published) and that they show relatively higher cytoplasmic fluorescence of FM4-64, which they use as a proxy for constitutively reduced endocytosis. For this correlation to be made, they need to show that a range of susceptible lines show 'reduced endocytosis' and that genetically diverse resistant lines consistently show elevated endocytosis.

Answer: Thank you very much for your constructive suggestions, it's very important for us to improve this manuscript. In the revised text, we provided new data on the endocytosis in root epidermal cells from several different susceptible rice lines (Figure S5).

2. In Figure 1, if sakuranetin production is induced by R protein activation, why would these NILs show constitutive reduced endocytosis?

Answer: Thank you very much for your comments. In the new version of the manuscript, we show that sakuranetin can decrease cellular endocytosis (Figure 2a-e). We also show that the rice NILs have higher sakuranetin levels (Figure S1, S2n), therefore, the rice NILs show constitutively reduced endocytosis (Figure S2a-m).

3. In Figure 1, what is the expression of NOMT gene expression in these lines (and in more susceptible lines), and what are the sakuranetin levels?

Answer: Thank you very much for your constructive suggestions! In the revised version of the manuscript, we provide new data showing sakuranetin levels in rice NILs (Figure S1, S2n) and susceptible lines (Figure S51).

4. Figure 1: I'm not sure what differences the images from m-w actually show?

Answer: Thank you very much for your comments. In the new Figure 1, the rice line LTH (Figure 1m) acted as a control, and the rice NILs (Figure 1n-w) showed enhanced fluorescence intensity on the root epidermal cells compared with the rice LTH (Figure 1x), illustrating that the rice NILs have reduced endocytosis compared with the control.

5. Figure S3 – sakuranetin treatment with 50 μ M provided resistance. Where was the dilution series to determine this concentration for use? How much is it induced anyway in resistant plants? What is the level of induction in LTH during infection? The authors only looked at one resistant line in this Figure – need to show all of the other resistant lines to demonstrate the correlation with increased sakuranetin production and NOMT expression and also include more susceptible lines to demonstrate this correlation holds up

Answer: Thank you very much for your critical comments, it's very important for us to improve this manuscript. In the new version of the manuscript, we provide data from rice lines LTH and Nipponbare treated with 25 μ M, 50 μ M and 100 μ M sakuranetin to induce rice resistance against rice blast (Figure S14). We also provided data of sakuranetin levels in *OsNOMT*-overexpressing rice lines (Figure S11c-d), *nomt-cc* mutant lines (Figure S12c-d) and susceptible lines (Figure S51).

6. Figure S5 describes overexpression of NOMT to increase sakuranetin levels. Where are the measurements that show the overexpression increases the phytoalexin production?

Answer: Thank you very much for your constructive suggestions! We have provided new data showing production of sakuranetin in the *OsNOMT*-overexpression rice lines (Figure S11c-d).

7. Figure 2 – overexpression of NOMT leads to reduced BFA body size, whereas NOMT KO is larger. What about resistance to *M. oryzae*? This is not included in this Figure but is instead included in Figure 5 at the end.

Answer: Thank you very much for your critical comments, it's really important for us to improve this text. In the revised document, the data pertaining to BFA body size and resistance in the *OsNOMT*-overexpression lines are shown in a single image (Figure 2m-t, aa-ac), and we have also provided data showing rice resistance in *nomt-cc* mutant lines (Figure 2ad-af).

8. Figure 3 – I fail to see the significance of this experiment. It shows that transformed rice expressing effector PWL2-GFP shows the effector at the PM and treatment with BFA and SAK results in less endocytosis of it (smaller BFA bodies). This is not a proxy for effector translocation from *M. oryzae*.

Answer: Thank you very much for your critical comments! In this new version of the text, we provide new data showing that rice lines expressing *PWL2-GFP* treated with BFA are able to internalize *PWL2-GFP* (Figure 3a-f); this demonstrates that *PWL2* can

be translocated into the cytoplasm as described in a previous report (Oliveira-Garcia, et al. 2023). Rice lines expressing *PWL2-GFP* and co-treated with sakuranetin and BFA showed smaller BFA bodies (Figure S3h-j), meaning that sakuranetin is able to decrease the endocytosis of effector *PWL2*. Further experiments demonstrated that sakuranetin treatment attenuated the translocation of the *M. oryzae* effector *AvrCo-39* into the rice cytoplasm (Figure S16).

9. Figure 4. The fact that use of CME inhibitor and clathrin mutation show the same phenotypes as SAK treatment does not mean that SAK targets CME. This is a correlation rather than a demonstration of genetic association.

Answer: Thank you very much for your comments! We agree that the TyrA23 treatment shows the correlation between sakuranetin and CME. In this revised text, we found that the sakuranetin-mediated decrease endocytosis of plasma membrane proteins was insensitive to the TyrA23 treatment, which is similar to the salicylic acid responses to TyrA23 treatment (Jiang et al., 2023). Therefore, in the new version of the manuscript, we have mentioned that “sakuranetin decreases CME of both PIN plasma membrane proteins and the fungal effector *PWL2*.” Furthermore, previous reports that the translocation of *M. oryzae* effector enters into rice cell via CME (Oliveira-Garcia, et al., 2023) further supports our theory that sakuranetin inhibits endocytosis of the effector *PWL2* via the CME pathway.

10. Figure 5. The new bit is the fact that overexpression of *NOMT* leads to resistance. This should be included in Figure 2. What does *NOMT* knockout do? The other components – *chc-cc* mutation and ES9 treatment repeat the recently published results on endocytosis of *M. oryzae* effectors (Oliveira-Garcia et al 2023) and are therefore not novel.

Answer: Thank you very much for your critical comments, it's very important for us to improve this manuscript. In our revised manuscript, the data investigating rice blast resistance in *OsNOMT*-overexpressing rice seedlings are shown in new images (Figure 2aa-ac), and we also provide new data regarding *nomt-cc* mutant resistance to rice blast (Figure 2ad-af).

The previous report on endocytosis of *M. oryzae* effectors is an impressive work and was published on March 28 (Oliveira-Garcia et al 2023). We claimed that our cellular experiments and resistance data were novel because we actually finished them prior to March 2023, however, we spent more time in improving the writing, and therefore delayed submission (our original text was first submitted to *Nature* on May 1 and transferred to *Nature Communications* on May 10). We are happy to know that the published data from Oliveira-Garcia et al, 2023 supports our results that the effector of *M. oryzae* enters into rice cells via CME pathway.

Minor comments

Introduction – when mentioning the endocytosis of *M. oryzae* effectors it is good to refer to the accompanying paper indicating that oomycete effectors also enter via endocytosis. This broadens the potential impact of your observation on sakuranetin.

Answer: We really appreciate for your constructive suggestions, it's very important for us to improve this manuscript. We have improved the writing and provide new references regarding the endocytosis of oomycetes and insects effectors.

Please ensure you have help to correct English errors throughout.

Answer: We really appreciated Reviewer 1 for her/his effort and time spent to help us improve this manuscript. We have carefully checked the text, and the writing has been improved by a native English speaker.

Reviewer #2 (Remarks to the Author):

General comment:

This study assessed the role of the phytoalexin sakuranetin inhibits endocytosis to enhance resistance to rice blast disease. Rice tissue treatment with sakuranetin inhibits endocytosis of plasma membrane localized proteins and fungal effectors translocation. These results were comparable to the treatment with Endosin9, a clathrin heavy chain inhibitor. Overexpression of the sakuranetin biosynthesis gene OsNOMT in rice also inhibits endocytosis and enhances disease resistance. The data analysis and statistics presented in this manuscript are appropriate. However, the authors must show individual data points in all graphs, precise P values and improve the quality of graphs and figures. The manuscript is moderately well written, and the outcomes should serve as references for fungal geneticists and plant pathologists. I support the publication of this manuscript in Nature Communications after proper corrections.

I have some comments that the authors should address.

Answer: We thank Reviewer 2 for her/his effort and time spent to help us to improve this manuscript. These constructive comments are very valuable to us and have allowed us to greatly improve this text. In the revised manuscript, we have improved the writing and quality of graphs and figures.

Specific comments:

The manuscript was missing line numbers, making it harder for the reviewers to evaluate the manuscript. The authors must resubmit the edited manuscript with numbered lines.

Answer: Thank you very much for your suggestion. In the revised text, we have provided line numbers.

Introduction:

“The plant immune response is critical to plant survival.” Please improve the sentence. “The plant immune system is essential to plant survival and adaptation to the environment”.

“are able to translocate into rice cells with CME”. It should be written: “are translocated into rice cell cytoplasm via CME”

Answer: Thank you very much for your suggestion. The writing of the new version of this manuscript has been improved throughout and has been checked by a native English speaker.

Figure 1: Authors must show individual data points in the graphs of this figure and in all graphs of the manuscript. Please present the graphs in solid colors. Use color-blind friendly colors for the figures. Why the disease index is negative? Use the free space to centralize the graphs L and X, and axel labels should be centralized. Graph legends are touching the axes labels. Size bars should be present on all figure panels. Please give the precise P values.

Answer: Thank you very much for your constructive suggestions, it's very important for us to improve this manuscript. In this revised manuscript, we have re-analyzed our data and improved all the figures.

Figure 2: Same comment regarding the quality of the graph. There is a great discrepancy in the font sizes for the axel's labels. Please give the precise P values.

Answer: Thank you very much for your constructive comments. We have improved the quality of figure 2.

Figure 3: Same comment regarding the quality of the graph. Please give the precise P values.

Answer: Thank you very much for your constructive comments. We have improved the quality of the graph and have provided precise *P* values.

Figure 4: Same comment regarding the quality of the graph. Please give the precise P values.

Answer: Thank you very much for your constructive comments. We have improved the quality of the graph and have provided precise *P* values.

Figure 5: Same comment regarding the quality of the graph. Size bars should be present on all figure panels. Please give the precise P values.

Answer: Thank you very much for your constructive comments. We have improved the quality of the graph and now show size bars and precise *P* values on all figure panels.

Endosidin9-17 is a more specific version of Endosidin9 lacking cytoplasmic acidification activity. Both ES9 and ES9-17 inhibit clathrin heavy chains.

Answer: Thank you very much for your critical comments, it's very important for us to improve this manuscript. In the revised text, we have provided new data for rice seedlings treated with ES9-17 (Figure 4g), and we found that both ES9 and ES9-17 treatment promoted rice resistance against rice blast (Figure 4d-i).

References: Please double check journal abbreviations and perhaps some appeared twice in two refence lists.

Answer: Thank you very much for your constructive comments. In this revised text, we have carefully checked the references.

Supplemental Figures: Same comments about graph quality, data points, scale bars, and legend resolution.

Answer: Thank you very much for your constructive comments. In this revised text, we have improved the quality of the graphs, data points and legend resolution, and have included scale bars.

Figure S6 is too small. Please improve it.

Answer: Thank you very much for your constructive comments. In this revised text, we have improved the figure quality.

Figure S8: “Bright” Is it bright field?

Answer: Thank you very much for your constructive comments. In this revised text, we have improved this figure and used “Bright field” to show the graph.

Reviewer #3 (Remarks to the Author):

The manuscript presented by Jiang and collaborators aims at a better understanding of immune response and resistance of rice to the pathogenic fungus *Magnaporthe grisea*, the causal agent of rice blast. The work deals with an intriguing subject related to the negative impact of the main rice phytoalexin, sakuranetin, on endocytosis of rice cells. The authors used nicely a combination of genetics, pharmacology and cell biology approaches on different rice lines to correlate the disease resistance with sakuranetin-triggered decrease of clathrin-mediated endocytosis (CME).

This is an innovative work but despite a significant set of complementary experiments (even if some of them sound outside the guideline) some conclusions remain elusive. The authors used different cultivars without clearly explaining why. The analysis and/or the presentation of the figures /quality of images (such blurred fluorescence and poor resolution) will for some of them to be improved.

Answer: We thank Reviewer 3 for her/his effort and time spent to help us to improve this manuscript. These constructive comments are very valuable to us and have allowed us to greatly improve this text. In the revised text, we have re-analyzed all the data and have improved the quality of the images.

Here are listed some detailed areas of concern.

Major points

The introduction indicated that endocytosis process is an important cellular event in plant microbe interaction, and its inhibition can attenuate the induction of plant defense. For instance, it is exemplified through M/D/PAMP-activated internalization by CME of membrane PRRs for degradation in the vacuole. This seems controversial compared with the present work, which showed that a defense component, the phytoalexin

sakuranetin, has a negative impact on the endocytosis. Neither in the introduction nor in the conclusion do the authors discuss the spatio-temporality of events. Indeed, PRR endocytosis is proposed as an early signalling event after signal perception, reflecting induced-endocytosis. In contrast, the decrease of constitutive endocytosis by sakuranetin can be considered as a late event requiring firstly the phytoalexin synthesis and secretion for plant resistance.

Answer: Thank you very much for your critical comments, they were very important for us to be able to improve this manuscript. In the revised text, we re-written the Introduction and Discussion sections.

The authors compared a susceptible line with a set of rice lines resistant to rice blast. Surprisingly, the disease was analysed on rice leaves and the endocytosis was carried out on root cells, using the endocytic marker FM4-64. Why the authors are not analysing the endocytosis of leaf cells, especially since sakuranetin level is higher (fig S3)?

Answer: Thank you very much for your critical comments and suggestion. We observed the endocytosis of rice leaf epidermal cells and found that the endocytic marker FM4-64 was not able to effectively stain leaf epidermal cells, and the presence of the chloroplast seriously affected our observations of the endosome (Please see below image). In this revised text, we have provided data for sakuranetin levels in the roots of rice NILs (Figure S2n).

The authors quantified the fluorescence of plasma membrane and cytoplasm to calculate a ratio between the two locations, testifying internalization if high. The method of quantification is not detailed and refers to Du et al 2013 where the inverse ratio was used (cytoplasm/PM) and where inverted black and white images were created, which are easier to analyse (notably when images are saturated as here). Quantification of the size of BFA bodies is a good complement of PM/cyt ratio, but no explanation of the characteristics of the bodies (or shape, number..) is given. The unit of the BFA bodies is missing in the ordinate legend.

Answer: Thank you very much for your critical comments, it's very important for us to improve this manuscript. In the new version of the manuscript, we have introduced the

method to quantify fluorescence intensity in detail, have improved the resulting images, and have provided the data for the number of BFA bodies (Figure 2k, t, z, S2m, S3g). Because we analyzed the relative size of the BFA bodies, we did not provide the unit of the BFA bodies in the ordinate legend, as is the case in previous reports (Jiang et al., 2023; Du et al., 2013)

Another concern is that the correlation between resistance and degree of inhibition of endocytosis is not clear (fig 1). Indeed, resistant lines present various level of disease index but this is not strictly correlated to the level of inhibition of endocytosis (e.g IRBLsh-S and IRBL9-W present a null disease index but the attenuation of endocytosis is not high at the contrary). Unfortunately, no data on sakuranetin concentration between resistant lines is given.

Answer: Thank you very much for your critical comments. In the revised manuscript, we conducted several new experiments and have provided new data for the disease index (Figure 1l) and fluorescence intensity (Figure 1x), and we also have provided data for the sakuranetin levels in the roots of rice NILs (Figure S2n).

The same statement could be done with overexpressed plants (fig2r) in which *OsNOMT* expression is not correlated with attenuation of endocytosis (fig S5, see plant #5 with low expression but endocytosis attenuation at the same level than the other overexpressors). Once again, roots have a lower expression of *OsNOMT*(which may indicate low levels of sakuratenin) than leaves but endocytosis was analysed in root cells.

Answer: Thank you very much for your constructive comments. In this revised text, we have conducted several new experiments and we found that the effect of sakuranetin decreased endocytosis via CME (Figure 3k-s). This suggests that the sakuranetin can effectively attenuate endocytosis in rice cells via the CME pathway even if there are differences in sakuranetin levels between the *OsNOMT*-overexpression lines. The expression of *OsNOMT* in the rice line #5 is not correlated with the effect of sakuranetin attenuated endocytosis, and it might be that the sakuranetin levels in line #5 is the lowest physiological concentration that leads to decreases the endocytosis in rice cells, therefore, the attenuated endocytosis in different *OsNOMT*-overexpression rice lines is at the same level. In this revised text, we have provided the levels of sakuranetin in the roots and leaves of *NOMT*-overexpressing lines (Figure S11c-d). As mentioned above, the dye FM4-64 does not stain leaf epidermal cells effectively, and we therefore observed cellular endocytosis in rice roots.

Cells treated with sakuranetin present a decrease (not an inhibition as written!) of the endocytosis and of the size of BFA bodies. It will be informative to test a dose-response of sakuranetin treatment. ◦

Answer: Thank you very much for your constructive suggestion. We conducted this experiment for the revised version of the manuscript. We found that rice seedlings treated with 10 μ M, 25 μ M and 50 μ M sakuranetin showed attenuated endocytosis

(Figure S6), and that this decrease in endocytosis is dependent on the dose of sakuranetin.

One main concern is that the sakuranetin is a lipophilic compound that has been shown to be incorporated in cell membranes (e.g see da Cruz Ramos Pires et al 2022, doi: 10.1016/j.colsurfb.2022.112546 112546, ISSN 0927-7765). We can therefore question whether this treatment could modify membrane rigidity and slow down the formation of endocytic vesicles.

Answer: Thank you very much for your critical comments, it's very important for us to improve this manuscript. In order to test whether the treatment modified membrane rigidity, we have developed a FRAP experiment and found that sakuranetin did not affect the fluidity of the plasma membrane (Figure S13). Furthermore, when rice lines with protoplast expressing marker genes *OsARA7-GFP*, *OsHDEL-GFP* and *OsSYP121-GFP* carrying GFP were treated with sakuranetin, we found that sakuranetin did not affect the movement of the endosome (Figure S10).

The ultrastructure images of rice epidermal cells (fig S7) is not really informative (low quality and contrast) and without quantification. A deeper analysis of such images will be attractive. For example, it seems that dense material is observed inside vacuoles of cells treated by sakuranetin. In such images, it will be also possible to scrutinized membranes or organelles (membrane waving, nascent endocytic structures, Golgi, Trans Golgi network, multivesicular bodies,...).

Answer: Thank you very much for your suggestion. For the revised version of the manuscript, we observed at least 10 ultrastructure images and did not find obvious differences in the phenotypes of different organelles, including the Golgi body, ER and multivesicular bodies in each sakuranetin treatment (Figure S9a-d), and therefore we did not provide any quantification data. The dense material observed inside vacuoles should be products from an oxidation reaction of lipids by osmic acid (OsO₄), and is not likely to be caused by sakuranetin. To further investigate the phenotypes of the organelles, protoplasts expressing *OsSYP121-GFP* labelled to TGN/EE were treated with sakuranetin, and we found that the TGN/EE could be internalized and did not show obvious differences compared to the control (Figure S9e-f).

The analysis of PWL2-GFP fate is very valuable to link decrease of endocytosis and decrease of the effector entry. The authors indicate that the fusion protein is localized in the plasma membrane, but I think it's better to employ "bound to membrane". The images of figure 3 do not present clear membrane resolution but the quantification of the size of BFA bodies seems convincing. Nevertheless, the figure title (and the conclusion) should be moderate as the amount of sakuranetin was only analysed in untransformed IRBL9-W(FigS3d) but not in the line expressing PWL2-GFP. Moreover, use "attenuates/decreases" instead of "inhibits".

Answer: Thank you very much for your critical comments. We have improved the text and the quality of Figure 3. We have provided data for the sakuranetin levels in rice lines IRBL9-W and LTH expressing *PWL2-GFP* (Figure S15).

The images of Fig S8 are not informative in that form, the red fluorescence is almost invisible. How quantification was achieved without running the risk of background noise? Same comment for figure S9. By the way the result of figure S9 (where the title should be change as it is not membrane protein but FM4-64 dye!) are difficult to apprehend and confirm only that sakuranetin has a strong impact on fungal cells. This is at the margin of the study.

Answer: Thank you very much for your critical comments, it's very important for us to improve this manuscript. In the revised text, we have provided new data for the fluorescence intensity of rice leaf sheaths (Figure S16) and have re-written the method for the measurement of fluorescence (Please see Supplemental Method section). We have removed the data pertaining to fungal spores stained with FM4-64.

Finally, the results with the PIN proteins complete the ones with FM4-64 but even if it show the decrease of endocytosis and inform of the nature of endocytic process, I wonder whether this is the best proteins to analyse, as it is not, in my opinion, a protein directly linked to immune response. Even if I may understand that it is a good tool.

Answer: Thank you very much for your critical comments! In the revised manuscript, we provide data showing that rice protoplast cells expressing the *OsSYP121-GFP* gene, which is involved in resistance to rice blast, also enhanced the localization of OsSYP121-GFP to the plasma membrane under sakuranetin treatment compared to the mock-treated controls (Figure S8). This shows that sakuranetin is also able to decrease endocytosis of proteins linked to the immune response.

Similarly, the work on root growth and gravity drowns out the main message of the paper.

Answer: Thank you very much for your constructive suggestion! In this revised text, we removed the data of root length and root growth angle of the *chc-cc* mutant.

Minor points

Many edits will have to be carried out. For instance:

- Significance of abbreviations are not always at the first citation.

Answer: Thank you very much for your constructive suggestion! In this revised text, we have carefully checked and re-written the abbreviations.

- Caption fig S1 should be “Endocytosis of plasma membrane...” instead of “endocytosis of plasma membrane proteins”. Indeed, FM4-64 reflects also the internalization of the lipid part of the membrane

Answer: Thank you very much for your critical comments! In this revised text, we have improved the writing for the caption Figure S2.

- Fig2r and Fig S2f: write for the ordinate axis “analyse of surface of PM” instead of “surface of PM protein internalization” as it is FM4-64 labelling!

Answer: Thank you very much for your critical comments! We have improved the description of the BFA body (Figure 2s, S3f).

- Write “endocytosis of root cells” (instead of “endocytosis of root”)

Answer: Thank you very much for your critical comments! We have improved the text describing the endocytosis of rice root cells.

- It is not a FM4-64 treatment but a FM4-64 labelling (use treatment for BFA or tyrphostin)

Answer: Thank you very much for your critical comments! We have improved the writing for the FM4-64 section.

In conclusion, despite significant works using many tools and lines and proposing an original hypothesis, I think that further experiments are needed to confirm and clarify the conclusions drawn. Authors need also to focus on certain results and not spread themselves too thin in order to get a clear message with greater impact.

Answer: We greatly appreciate Reviewer 3 for her/his effort and time spent to help us to improve this manuscript, her/his comments and suggestions are very valuable for us to improve this text. In this revised text, we have provided new data and also removed some data to confirm our hypothesis. We really hope this text could be recommended for publication in *Nature Communications*.

We greatly appreciate reviewers’ work and sincerely hope that the reviewers will reconsider and accept this manuscript after our extensive revisions, and will recommend its publication in *Nature Communications*.

With kind regards,

Yunlong Du (corresponding author)

Dr. Yunlong Du, Professor, College of Plant Protection, Yunnan Agricultural University, 650201 Kunming, China.

Phone: +86 18206799459.

E-mail: yunlongdu@aliyun.com

REFERENCES:

Du, Y et al. Salicylic acid interferes with clathrin-mediated endocytic protein trafficking. PNAS. 110(19):7946-7951, 2013.

Jiang, L et al. Salicylic acid inhibits rice endocytic protein trafficking mediated by OsPIN3t and clathrin to affect root growth. The Plant Journal. 115(1):155-174, 2023. DOI: 10.1111/tpj.16218. 2023.

Oliveira-Garcia, E. et al. Clathrin-mediated endocytosis facilitates the internalization of *Magnaporthe oryzae* effectors into rice cells. *The Plant cell*, 35, 2527-2551, 2023.

Reviewer #2 (Remarks to the Author):

General comment:

This manuscript has elucidated the role of rice phytoalexin "sakuranetin" in inhibiting endocytosis of plasma membrane localized proteins and fungal effectors translocation. The authors have done an excellent job incorporating the reviewers' comments into the manuscript. Data analysis and statistics are appropriate. The manuscript is now well written, and the outcomes should serve as references for fungal geneticists and plant pathologists. I support the publication of this manuscript in Nature Communications after minor corrections.

I have some comments that the authors should address.

Specific comments:

Figures. The confocal images are high quality. I would suggest changing the red to magenta to facilitate the interpretation of color-blind people. You can also check if the colors are suitable for color-blind people using free software, such as EnChroma, EyeQue etc.

Graphs could be presented in colors. The graphs in Figure 2 I, J and K contain vertical and horizontal line patterns that make harder the interpretation. Colors or gray scale would facilitate their interpretation.

Figure 4 legend, line 578: It is missing a period after endosidin9-17.

Great job!

Reviewer #3 (Remarks to the Author):

The new version of the manuscript proposed by Jiang and colleagues has been improved, both by carrying out new experiments and more detailed analyses / quality presentation of the results, and by modifying the text. This indicates a huge effort from the authors who have considered the comments and criticisms made.

However, the text needs to be carefully checked. Some concerns remain to be addressed, corrected or discussed. For example:

Line 29: "effects of R gene products" instead of "effect of the genes..?"

Line 40 "Pharmacological manipulation using clathrin-mediated endocytic mechanism" is not clear, change to "Pharmacological manipulation of CME..."

Line 42: "...by other independent means" is not really discussed in the manuscript

Line 56 and along the text (and figure legends): write "near-isogenic lines" instead of "near isogenic lines"

Line 82: the sentence "...phytoalexin sakuranetin can enhance plant immunity by regulating cellular endocytosis" is confusing because phytoalexin synthesis is considered to be a crucial part of the innate immune response of plants! Thus, phytoalexin is part of the immune response but must be considered as participating in resistance. A more appropriate sentence could therefore be corrected as follows: "...phytoalexin sakuranetin may enhance plant resistance by down-regulating cellular endocytosis".

Line 93: replace with "...by inoculating them with the Guy11 strain of *M. oryzae*". (without capitals)

Line 94: I think the usual spelling is "Lijiangxintuanheigu (LTH)" without separate parts.

Line 98: more appropriate "...with FM4-64, the established endocytic fluorescent tracer,...".

Line 100: write "the PM fluorescence was significantly increased..." or more logically "the cytoplasm fluorescence was significantly decreased..." because the figures show the cyt/PM ratio in this new version.

Line 102 "...analysed FM4-64 endocytosis in NILs... thus, remove in line 103 the sentence: "Cells were labelled..."

Line 106: you cannot write "that this reduction was accompanied by higher levels of sakuranetin". It is indeed true with leaf content but FigS2n shows "higher" root content (but below 1ng/g FW)

for only 6 NILs among the 10. Be cautious about this statement. Same for line 123 for susceptible lines, the term "accompanied" should be at least changed.

Line 152: "role in PM localization" of what?

Line 171: to correct: HDEL is not a receptor but an ER retention signal

Line 204: "...impact of sakuranetin on endocytosis does not impair PM fluidity" this sentence is confusing as the opposite is also true: the endocytosis attenuated by sakuranetin is not due to a change in membrane fluidity!

Line 217: explain what is MoPot2, if defense gene write "MoPt2 expression"

Line 262: "similar like in case of PIN proteins", proper English? Better as "similar to that found for PINs"

Line 267: mention CME in the title

Line 289: the statement "...we discovered a mechanism underlying sakuranetin's role ...". this assertion is a little strong because the regulation of the endocytosis process (and particularly CME) has been indeed discovered but the mechanism (i.e. how this happens) has not been really explained in this study.

Line 293: PM proteins

Line 296: "...endocytic uptake of a fluorescent tracer, correlated with higher endogenous levels of sakuranetin..

Line 318: explain the nature of signal (e.i MAMP/DAMP..); "reflecting ligand-induced endocytosis"

Lin 320: "and then its secretion"

Line 321: "...sakuranetin down-regulating CME provide a ..."

Line 323: "identification of the mechanism underlying sakuranetin effect on CME.."

Line 322: "other plant species against diverse pathogen infections"

I also suggest for supplemental figures S2 and S6 to write the letters in another colour than red (such yellow) because it is hard to see them.

Dear Reviewers,

In the revised manuscript (ID: NCOMMS-23-20536A-Z), we have carefully addressed all comments in full.

We greatly appreciate the effort, help and constructive suggestions of Reviewers #2 and #3, which have greatly helped us to improve this manuscript.

As you will see below, I have responded to the Reviewers' comments point-by-point.

With kind regards,

Yunlong Du (corresponding author)

Dr. Yunlong Du, Professor, College of Plant Protection, Yunnan Agricultural University, 650201 Kunming, China.

Phone: +86 18206799459.

E-mail: yunlongdu@aliyun.com

REVIEWERS' COMMENTS

Reviewer #2 (Remarks to the Author):

General comment:

This manuscript has elucidated the role of rice phytoalexin “sakuranetin” in inhibiting endocytosis of plasma membrane localized proteins and fungal effectors translocation. The authors have done an excellent job incorporating the reviewers' comments into the manuscript. Data analysis and statistics are appropriate. The manuscript is now well written, and the outcomes should serve as references for fungal geneticists and plant pathologists. I support the publication of this manuscript in Nature Communications after minor corrections.

Answer: We thank Reviewer #2 for her/his effort and time spent in helping us to improve this manuscript. These comments are very valuable to us and have allowed us to improve the text.

I have some comments that the authors should address.

Specific comments:

Figures. The confocal images are high quality. I would suggest changing the red to magenta to facilitate the interpretation of color-blind people. You can also check if the colors are suitable for color-blind people using free software, such as EnChroma, EyeQue etc.

Answer: Thank you very much for your constructive suggestion; it is very important for us to improve this manuscript. In this revised text, we have carefully checked the figures and changed the red to magenta in Supplementary Figure 16.

Graphs could be presented in colors. The graphs in Figure 2 I, J and K contain vertical and horizontal line patterns that make harder the interpretation. Colors or gray scale would facilitate their interpretation.

Answer: In this revised text, we have changed the graphs in Figure 2E, J and K to gray scale.

Figure 4 legend, line 578: It is missing a period after endosidin9-17.

Answer: In this revised text, we have carefully checked and improved the writing in the whole document.

Great job!

Answer: Thank you very much for your comments!

Reviewer #3 (Remarks to the Author):

The new version of the manuscript proposed by Jiang and colleagues has been improved, both by carrying out new experiments and more detailed analyses / quality presentation

of the results, and by modifying the text. This indicates a huge effort from the authors who have considered the comments and criticisms made.

Answer: We thank Reviewer #3 for her/his effort and time spent in helping us to improve this manuscript. These constructive comments are very valuable to us and have allowed us to greatly improve the text.

However, the text needs to be carefully checked. Some concerns remain to be addressed, corrected or discussed. For example:

Line 29: “effects of R gene products” instead of “effect of the genes..”?

Answer: In the revised text, we have improved the writing throughout the document.

Line 40 “Pharmacological manipulation using clathrin-mediated endocytic mechanism” is not clear, change to “Pharmacological manipulation of CME...”

Answer: Thank you very much for your constructive suggestions, it’s very important for us to improve this manuscript. In the revised text, we have altered this part to “Pharmacological manipulation of the clathrin-mediated endocytic (CME) mechanism suggested that this pathway is targeted by sakuranetin.”

Line 42: “..by other independent means” is not really discussed in the manuscript

Answer: You are right; we really did not discuss “other independent means” in this study. In this revised version, we have altered the text to “...attenuation of CME by sakuranetin is sufficient to convey resistance against rice blast.”

Line 56 and along the text (and figure legends): write “near-isogenic lines” instead of “near isogenic lines”

Answer: Done

Line 82: the sentence "...phytoalexin sakurantin can enhance plant immunity by regulating cellular endocytosis" is confusing because phytoalexin synthesis is considered to be a crucial part of the innate immune response of plants! Thus,

phytoalexin is part of the immune response but must be considered as participating in resistance. A more appropriate sentence could therefore be corrected as follows: "...phytoalexin sakurantin may enhance plant resistance by down-regulating cellular endocytosis".

Answer: Thank you very much for your constructive suggestions, it's very important for us to improve this manuscript. In this revised text, we have altered the text according to your suggestion.

Line 93: replace with "...by inoculating them with the Guy11 strain of *M. oryzae*".
(without capitals)

Answer: Done.

Line 94: I think the usual spelling is "Lijiangxintuanheigu (LTH)" without separate parts.

Answer: Done.

Line 98: more appropriate "...with FM4-64, the established endocytic fluorescent tracer,...".

Answer: Done.

Line 100: write "the PM fluorescence was significantly increased..." or more logically "the cytoplasm fluorescence was significantly decreased..." because the figures show the cyt/PM ratio in this new version.

Answer: Thank you very much for your constructive suggestion. We have altered the text to "the cytoplasm fluorescence was significantly decreased in the root epidermal cells of all the tested rice NILs (Figure 1n-x)."

Line 102 "...analysed FM4-64 endocytosis in NILs... thus, remove in line 103 the sentence: "Cells were labelled..."

Answer: Done.

Line 106: you cannot write “that this reduction was accompanied by higher levels of sakuranetin”. It is indeed true with leaf content but FigS2n shows "higher" root content (but below 1ng/g FW) for only 6 NILs among the 10. Be cautious about this statement. Same for line 123 for susceptible lines, the term “accompanied” should be at least changed.

Answer: Thank you very much for your critical comments! In the revised text, we have changed the text to “and that this reduction was correlated with higher levels of sakuranetin (Supplementary Figure 2a-n).” and “and this decrease was shown in all cases by lower sakuranetin levels than the control (Supplementary Figure 5l).”

Line 152: “role in PM localization” of what?

Answer: In the revised text, we have changed the text to “which localizes in the PM...”

Line 171: to correct: HDEL is not a receptor but an ER retention signal

Answer: Thank you very much for your critical comments! In the revised text, we have changed the text to “the endoplasmic reticulum (ER) labeled with the ER retention signal protein (HDEL)”.

Line 204: “..impact of sakuranetin on endocytosis does not impair PM fluidity” this sentence is confusing as the opposite is also true: the endocytosis attenuated by sakuranetin is not due to a change in membrane fluidity!

Answer: Thank you very much for your critical comments! In the revised text, we have changed the text to “...sakuranetin treatment does not change PM fluidity”.

Line 217: explain what is MoPot2, if defense gene write "MoPt2 expression"

Answer: In the revised text, we have altered the text to “...*MoPot2*, which is a transposon used to detect the genetic diversity of *M. oryzae*.”

Line 262: “similar like in case of PIN proteins”, proper English? Better as “similar to that found for PINs”

Answer: In this revised text, we have altered the text according to your suggestion.

Line 267: mention CME in the title

Answer: Done

Line 289: the statement “..we discovered a mechanism underlying sakuranetin’s role ..”. this assertion is a little strong because the regulation of the endocytosis process (and particularly CME) has been indeed discovered but the mechanism (i.e. how this happens) has not been really explained in this study.

Answer: Thank you very much for your critical comments! In the revised text, we have altered the text to “...we showed a mechanism underlying sakuranetin’s role in defense against rice blast.”

Line 293: PM proteins

Answer: Done.

Line 296: “...endocytic uptake of a fluorescent tracer, correlated with higher endogenous levels of sakuranetin..

Answer: Done.

Line 318: explain the nature of signal (e.i MAMP/DAMP...); “reflecting ligand-induced endocytosis”

Answer: Thank you very much for your critical comments. In the revised text, we have altered the text to “Endocytosis of the pattern recognition receptor (PRR) of MAMP (Microbe-associated molecular pattern) is proposed as an early signaling event after signal perception.”

Lin 320: “and then its secretion”

Answer: Done.

Line 321: “..sakuranetin down-regulating CME provide a ...”

Answer: Done.

Line 323: “identification of the mechanism underlying sakuranetin effect on CME..”

Answer: Done.

Line 322: “other plant species against diverse pathogen infections”

Answer: Done.

I also suggest for supplemental figures S2 and S6 to write the letters in another colour than red (such yellow) because it is hard to see them.

Answer: Thank you very much for your comments. In this revised text, we have changed the colour of the letters in Figure S2 and S6.